# SOD3 suppresses early cellular immune responses to parasite infection

Qilong Li [1,2,7], Kunying Lv [1,2,7], Ning Jiang [1,2,7], Tong Liu[1,2], Nan Hou [3], Liying Yu [1,2], Yixin Yang [1,2], Anni Feng [1,2], Yiwei Zhang [1,2], Ziwei Su [1,2], Xiaoyu Sang [1,2], Ying Feng [1,2], Ran Chen [1,2], Wenyue Xu [4], Liwang Cui [5], Yaming Cao[6] & Qijun Chen [1,2] ✉

Host immune responses are tightly controlled by various immune factors during infection, and protozoan parasites also manipulate the immune system to evade surveillance, leading to an evolutionary arms race in host–pathogen interactions; however, the underlying mechanisms are not fully understood. We observed that the level of superoxide dismutase 3 (SOD3) was significantly elevated in both *Plasmodium falciparum* malaria patients and mice infected with four parasite species. SOD3-deficient mice had a substantially longer survival time and lower parasitemia than control mice after infection, whereas SOD3-overexpressing mice were much more vulnerable to parasite infection. We revealed that SOD3, secreted from activated neutrophils, bound to T cells, suppressed the interleukin-2 expression and concomitant interferon-gamma responses crucial for parasite clearance. Overall, our findings expose active fronts in the arms race between the parasites and host immune system and provide insights into the roles of SOD3 in shaping host innate immune responses to parasite infection.

Host immune responses are tightly controlled by various immune factors during infection[1,2]. Pathogens are known to hijack host factors[3] or express an array of virulence factors[4–7] that aim to overcome host immune defenses to achieve successful proliferation and dissemination. Host genetic and immune factors (HIFs) are the two major types of factors that are exploited by pathogens during infection[8,9]. The reactive oxygen species (ROS) are one types of the innate responses upon pathogen invasion[10]. However, ROS is not only detrimental to the invaded pathogens, but also harmful to the host cells. The host cells rely on the superoxide dismutases (SODs, including SOD1, SOD2 and SOD3) to scavenge the extra amount of ROS and the maintenance of

the homeostasis. It is known that *Plasmodia* increased a substantial amount of SOD activity from host in the infection[11]. However, a fundamental challenge in studying the HIF-pathogen interactions is to understand the host derive-redox signaling that facilitates pathogen evasion of host immune responses. Superoxide dismutase 3 (SOD3) is the only SOD-like enzyme secreted from the host cells to the extracellular space that scavenges substantial amounts of ROS to protect the host from oxidative stress during infection[12,13]. Additionally, SOD1, primarily functions in the cytoplasm, has been reported to serve as an indicator of disease severity in individuals with various clinical manifestations of *vivax* malaria[14]. SOD2, mainly localized in the

[1]Key Laboratory of Livestock Infectious Diseases, Ministry of Education, and Key Laboratory of Ruminant Infectious Disease Prevention and Control (East), Ministry of Agriculture and Rural Afairs, College of Animal Science and Veterinary Medicine, Shenyang Agricultural University, 120 Dongling Road, Shenyang 110866, China. [2]Research Unit for Pathogenic Mechanisms of Zoonotic Parasites, Chinese Academy of Medical Sciences, 120 Dongling Road, Shenyang 110866, China. [3]NHC Key Laboratory of Systems Biology of Pathogens, Institute of Pathogen Biology, Chinese Academy of Medical Sciences & Peking Union Medical College, Beijing, China. [4]Department of Pathogenic Biology, Army Medical University (Third Military Medical University), Chongqing 400038, China. [5]Department of Internal Medicine, Morsani College of Medicine, University of South Florida, Tampa, FL, USA. [6]Department of Immunology, China Medical University, 77 Puhe Road, Shenyang 110122, China. [7]These authors contributed equally: Qilong Li, Kunying Lv, Ning Jiang. ✉ e-mail: qijunchen759@syau.edu.cn

mitochondrial lumen, has been found to be notably upregulated in pre-asymptomatic malaria patients from Cameroon[15]. Although the roles of SOD1 and SOD2 in malaria are well characterized[14,15], almost nothing is known about the contribution of SOD3 to host immune responses beyond ROS scavenging.

Here, we demonstrated that SOD3 mainly secreted by neutrophils could directly bind T cells and suppress its IL-2 production, and consequently reduced the recruitment of IFN-γ producing T cells in the responses to invading parasites. Our results revealed an essential role of SOD3 in responses to parasite invasion and targeting SOD3 may be an alternative way to enhance host protective responses and reduce disease severity in acute infection.

## Results

### The expression of SOD3 was elevated in malaria patients and protozoa infected mice

We compared the expression of SOD3 in mice infected with lethal *Plasmodium y. yoelii* YM at different time points post-infection. Contrast to that in uninfected mice, the expression of SOD3 in the splenocytes was significantly elevated when parasites were detectable in the blood (Fig. 1a and Supplementary Fig. S1a, b). Upregulation of serum SOD3 in WT mice after infection with *P. berghei* ANKA (another

rodent malaria parasite that is also lethal to the host, Fig. 1b) and *P. y. yoelii* YM (Supplementary Fig. S1c) was also observed, and that elevation of serum SOD3 was associated with increased parasitemia. We further quantitatively analyzed SOD3 in the sera of patients infected with *P. falciparum vs.* that of healthy donors. The patients with a first malaria episode had significantly higher levels of SOD3 than either individual with chronic malaria or healthy donors from nonmalaria epidemic areas (Fig. 1c).

To determine whether elevated SOD3 expression is a general phenomenon associated with protozoan infection. We extended the study by measuring the SOD3 levels in mice infected with either *Trypanosoma brucei brucei* Lister 427 or *Toxoplasma gondii* RH, and it turned out to be the same as that observed in mice infected by the two *Plasmodium* species (Supplementary Fig. S1d, e). Therefore, we hypothesize that SOD3 is essentially associated with host vulnerability to protozoan infection.

To test this hypothesis, we used SOD3 knockout (SOD3[−/−]) mice, in which SOD3 was not detectable in the sera by LC–MS/MS analysis (Supplementary Fig. S1f, g). We further compared the vulnerability of the SOD3[−/−] mice and their WT littermates to infection of lethal *Plasmodium* species. To our surprise, the SOD3[−/−] mice showed significant resistance, with substantially extended survival time, to the infection

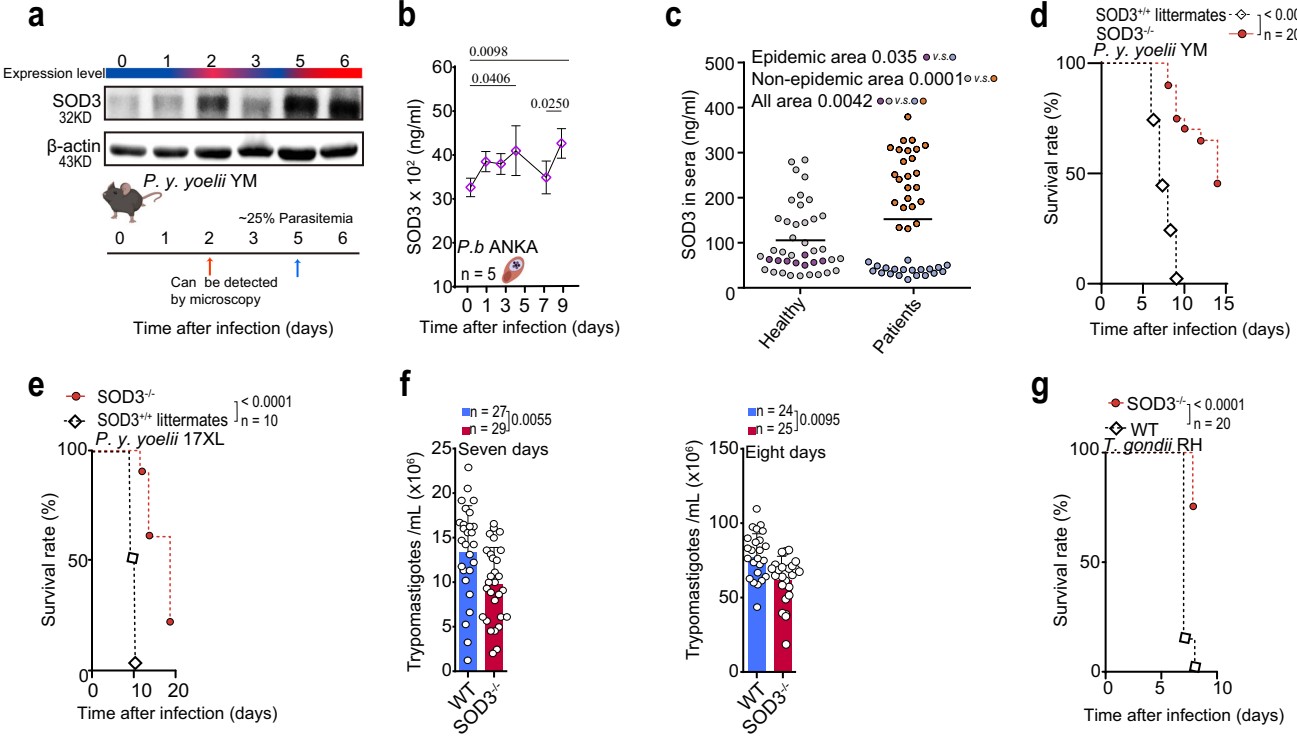

**Fig. 1 | SOD3 is a host factor associated with host vulnerability to parasite infection. a** Progressively increased expression of SOD3 was observed during *P. y. yoelii* YM infection. The mouse illustration was created with http://BioRender.com (publishing license: CZ26UIGFUP). **b** Serum SOD3 expression was elevated in mice after infection with *P. berghei* ANKA, with peaks observed after 5 days and 9 days. Five biological replicates were included in the analysis; healthy controls were three biological replicates. Statistical significance tested by one-way ANOVA, and multiple comparisons to each day were corrected using Dunnet's method. WT mice of average survival time was 9 days from the time of challenge. **c** SOD3 in the sera of *P. falciparum*–infected patients from the China-Myanmar border (endemic areas, n = 20), other endemic areas (cases were travel-related, n = 24) and healthy donors (endemic and nonendemic areas, n = 41) as measured by enzyme-linked immunosorbent assay (ELISA). SOD3 levels were significantly elevated in malaria patients from all areas compared with corresponding healthy donors. Two-tailed student's *t* test was used to test for significant differences between two groups. **d** SOD3[−/−] mice

(n = 20) showed significant extended survival time than the WT littermates (n = 20) after *P. y. yoelii* YM infection. Survived mice in SOD3[−/−] group were humanely euthanized accordingly to the Institutional Animal Care and Use Committee-approved criteria. **e** Knockout of SOD3 extended the survival of *P. y. yoelii* 17XL-infected mice, n = 10. The survived mice in SOD3[−/−] group were humanely euthanized accordingly to approved criteria. **f** Knockout of SOD3 decreased parasitemia in both seven and eight days after *T. b. brucei* infection. Two-tailed student's *t* test was used to test for significant differences between two groups. **g** Knockout of SOD3 extended the survival of *T. gondii*-infected mice. The survived mice in SOD3[−/−] group were humanely euthanized accordingly to approved criteria. Kaplan–Meier survival curves were calculated using the survival time for each mouse in all groups, and significance was determined by the log-rank test. For d-f, n indicates mouse numbers in graphs. Histograms present the mean ± SD. Source data are provided as a Source Data file.

of either *P. y. yoelii* YM strain or *P. y. yoelii* 17XL strain, which are both lethal to mice (Fig. 1d, e). To further determine whether SOD3 is generally associated with host vulnerability to protozoan infection, SOD3[-/-] and WT mice were respectively infected with two other protozoan parasites, including *Trypanosoma brucei brucei* Lister 427 and *Toxoplasma gondii* (RH strain). SOD3[-/-] mice also showed significant resistance to the two parasite species compared to the WT mice (Fig. 1f, g). Therefore, these results suggested that SOD3 is essentially associated with host vulnerability to protozoa infection.

### The expression of SOD3 is associated with experimental cerebral malaria (ECM)

Cerebral malaria is a severe complication which is caused by the sequestration on *P. falciparum*-infected erythrocytes in the cerebral microvasculature[16]. Cerebral pathogenesis of *P. falciparum* malaria was experimentally modeled with *P. berghei* ANKA-infected C57BL/6 mice[17]. Here, we found that the expression of SOD3 was higher in the brains of *P. berghei* ANKA-infected mice than that in the uninfected WT mice as illustrated in the immunohistochemistry analysis (Supplementary Fig. S2a). We next investigated whether SOD3 expression is associated with *P. berghei* ANKA-induced ECM. We used a transgenic luciferase-expressing line, *P. berghei* ANKA (*Pb*A luc), to infect SOD3[-/-], SOD3-overexpressing (SOD3[o/e], isogenic to C57BL/6) and WT mice, and quantified the luminescence via in vivo luminescence imaging. More severe ECM were observed in the SOD3[o/e] mice (Fig. 2a). The sequestration of infected red blood cells (iRBCs) in the brain of SOD3[o/e] mice

was approximately 1.5-fold more compared to that of WT mice (Fig. 2b). In contrast, the amount of sequestered iRBCs in the brain of SOD3[-/-] mice was significantly lower than that in WT mice (Fig. 2a, b), and the SOD3[-/-] mice survived substantially longed after *P. berghei* ANKA infection (Fig. 2c), while both SOD3[o/e] and WT mice succumbed to death due to ECM at the same time after infection (Fig. 2c, d). Further, the malaria-associated pathologies in the SOD3[-/-] mice was alleviated compared to WT mice (Supplementary Fig. S2b, c). Importantly, intravenously administration of recombinant SOD3 in SOD3[-/-] mice reversed their resistance to *P. berghei* ANKA infection (Fig. 2e). Furthermore, *P. berghei* ANKA isolated from SOD3[-/-] mice and WT mice displayed similar infectivity in naïve mice, suggesting that SOD3 had no direct effect on the parasites (Supplementary Fig. S3d and e). All data indicate that SOD3 is an important factor associated with host vulnerability to protozoa infection.

### SOD3 suppresses the function of T cells in parasite infection

To investigate whether SOD3 can directly bind and regulate immune cells, we conducted cell binding experiments using the recombinant SOD3 protein (rSOD3). Result showed that the rSOD3 directly bound to T cells and NK cells (Fig. 3a). Early burst of IFN-γ production is associated with protection to *P. berghei* ANKA infection. Intriguingly, a significant increase of IFN-γ in early stage was detected in the SOD3[-/-] mice, indicating a negative correlation between SOD3 and IFN-γ production, which was even more significant during early infection (Fig. 3b). The results of immunohistochemical staining of spleen tissue

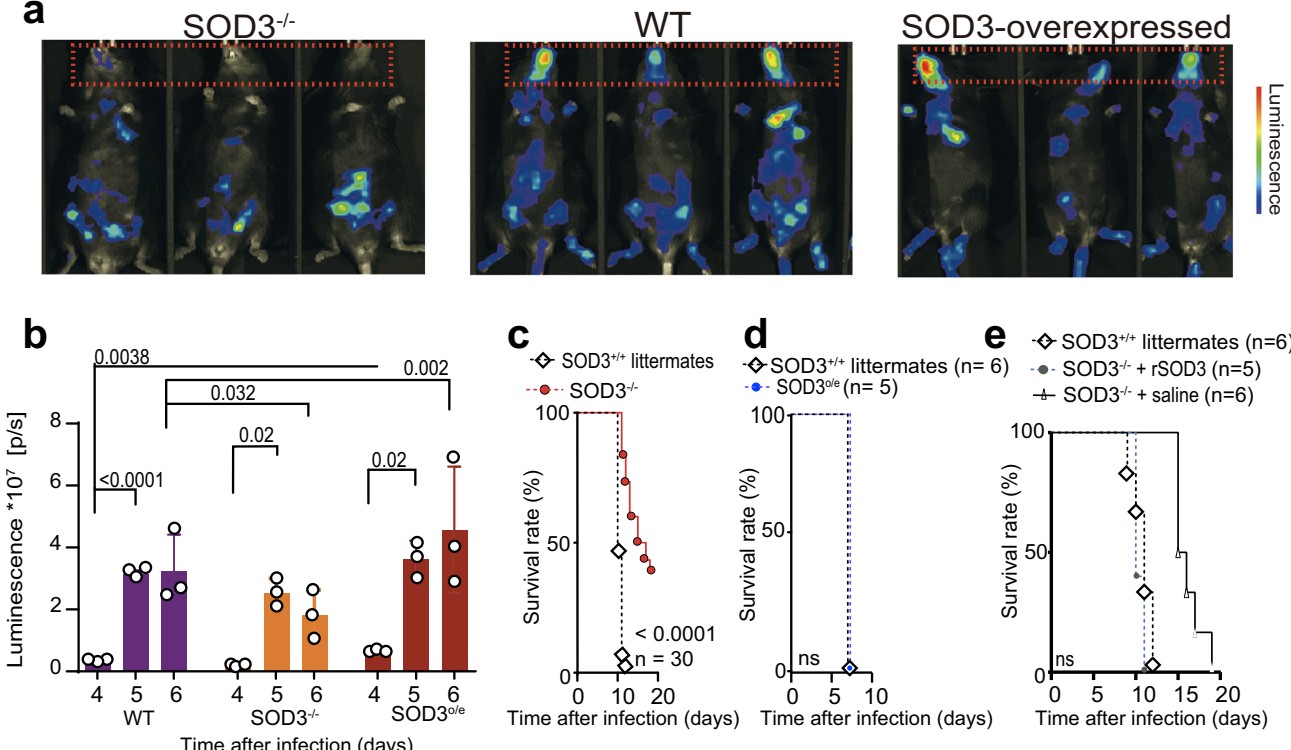

**Fig. 2 | Elevated SOD3 expression is associated with the severity and experimental cerebral malaria (ECM) in murine models. a, b** SOD3 is associated with cerebral malaria in WT mice. The accumulation of *P. berghei* iRBCs was not obvious in the mouse brains of SOD3[-/-] mice. However, this phenomenon was reversed in both littermate (WT) and SOD3[o/e] mice after infection, which showed severe sequestration of iRBCs in the brains. Overexpression of SOD3 also resulted in increased parasitemia, as reflected by increased luminescence signals in the body. The luminescence images of the mice were recorded with an AniView600 multimode in vivo animal imaging system, *n* = 3. Statistical tests were two-sided, and Tukey corrected for multiple comparisons. **c** SOD3[-/-] mice displayed extended

survival after *P. berghei* infection. The survived mice in SOD3[-/-] group were humanely euthanized accordingly to approved criteria. **d** SOD3[o/e] mice and control mice showed similar vulnerability to the parasite infection. Five biological replicates were used for analysis. Kaplan–Meier survival curves were calculated using the survival time for each mouse in all groups, and significance was determined by the log-rank test. **e** Administration of recombinant SOD3 increased the sensitivity of the SOD3[-/-] mice to *P. berghei* infection. For (**c–e**), n indicates mouse numbers in graphs. For c and e, Kaplan-Meier survival curve *p*-values were performed using Log rank Mantel-COX test. Histograms present the mean ± SD. Source data are provided as a Source Data file.

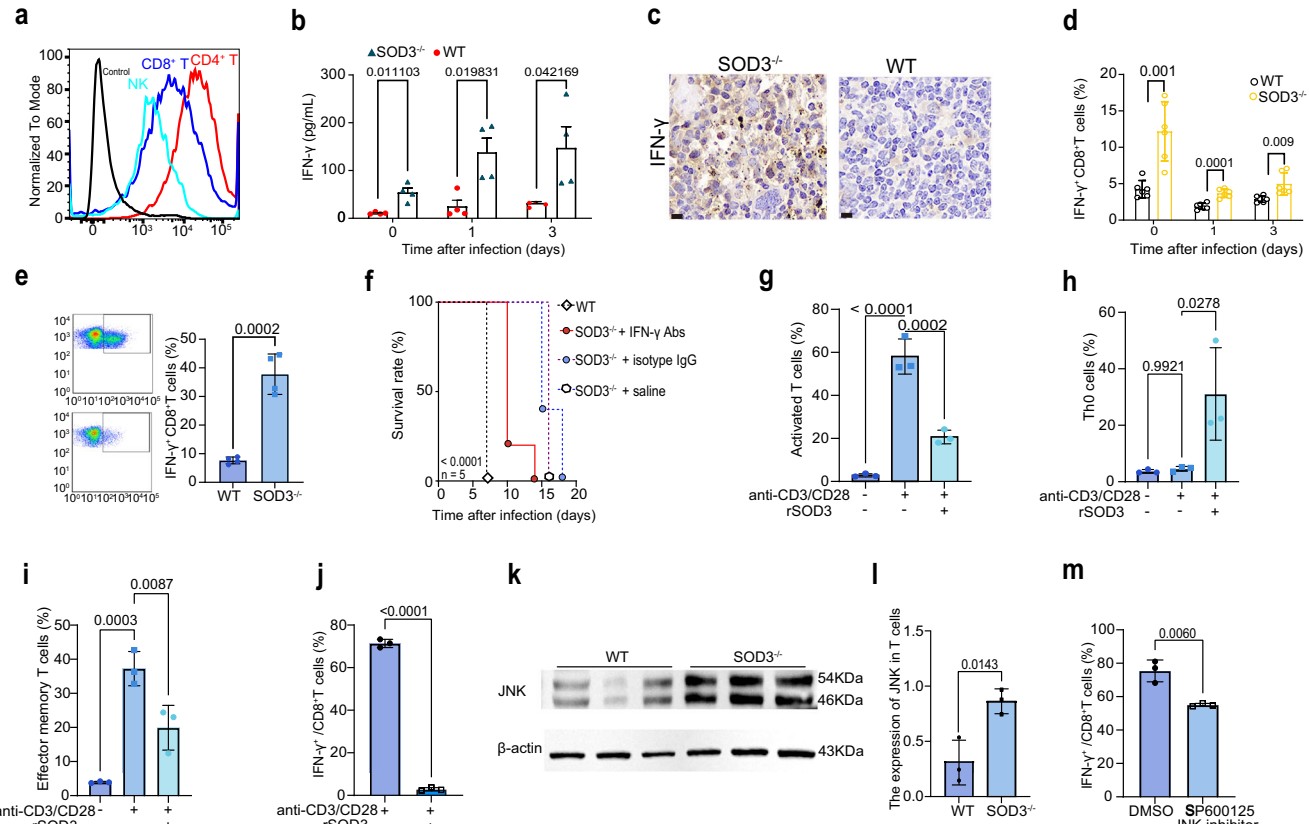

**Fig. 3 | SOD3 bound T cells and suppressed the IFN-γ response to parasite infection. a** SOD3 can bind both T cells and NK cells. Black line indicates the blank control, and the gray line represents omission of the primary antibody as the negative control. The Alexa Fluor™ 488-conjugated goat anti-Rabbit IgG (H + L) secondary antibody was used to recognize the primary antibody. The control sample was stained with a non-specific rabbit IgG as the primary antibody and AF488-goat anti-rabbit IgG secondary antibody. **b** Depletion of SOD3 in mice rescued the IFN-γ production in the sera (*n* = 4). **c** The expression of IFN-γ protein in the splenic cells of SOD3⁻/⁻ mice was much more than that of WT mice after infection illustrated by immunohistochemistry. Scale bar =20 μm. **d, e** The proportion of IFN-γ-expressing CD8⁺ T cells was significantly enhanced in SOD3⁻/⁻ mice infected with *P. y. yoelii* (yellow bar). *n* = 6 (**d**); *n* = 3 (**e**). **f** Depletion of IFN-γ in mice resulted in early death in SOD3⁻/⁻ mice after *P. berghei* ANKA infection. Five biological replicates were included in the analysis. Kaplan–Meier survival curves were

calculated using the survival time for each mouse in all groups, and significance was determined by the log-rank test. **g** SOD3 inhibited T cell activation (*n* = 3). **h, i** SOD3 inhibited the differentiation of Th0 cells (naïve T cells) into effector cells in in vitro experiments (*n* = 3). **j** The proportion of IFN-γ-expressing CD8⁺ T cells upon stimulation with SOD3 was significantly reduced in in vitro experiments (*n* = 3). **k, l** Western blot and quantitative for JNK and β-actin in cell lysates obtained from T cells sorted from SOD3⁻/⁻ and WT mice after infection. The expression of JNK in T cells was significantly lower in WT mice than that in the SOD3⁻/⁻ mice. Representative immunoblot using antibodies against β-actin and JNK1/2/3 (*n* = 3). **m** Treatment with SP600125, a selective JNK inhibitor, led to a decrease in the proportion of IFN-γ-expressing CD8⁺ T cells (*n* = 3). *t*-test two-sided was used to compare change in (**b, d, e, g–j, l, m**). Histograms present the mean ± SD. Source data are provided as a Source Data file.

using IFN-γ antibody confirmed the aforementioned findings (Fig. 3c). It is well known that CD8⁺ T cells and NK cells are major sources of IFN-γ, which play a pivotal role in the control of invading pathogens[18]. Here, the IFN-γ-expressing CD8⁺ T cell subsets were also expanded following *P. berghei* (Fig. 3d) or *P. y. yoelii* (Fig. 3e) infection in SOD3⁻/⁻ mice, suggesting that SOD3 may inhibit the activity of T cells and depletion of SOD3 would result in expansion of these IFN-γ-expressing T, NK and NKT cell subsets (Supplementary Fig. S3a–e). As all three major types of IFN-γ-producing T, NK and NKT subpopulations express IFN-γ at steady state (Supplementary Fig. S3f–j), which indicated that SOD3 may be an add-on factor in modulation of IFN-γ expression. Moreover, IFN-γ depletion led to a significant reduction in survival time of 80% SOD3⁻/⁻ mice after infection, confirming that IFN-γ plays a protective role in SOD3⁻/⁻ mice (Fig. 3f).

To investigate whether SOD3 can directly inhibit the expansion of IFN-γ-producing T cell subpopulations, we performed T-cell activation and differentiation assay in vitro. rSOD3 could directly suppress T cell activation (Fig. 3g) and inhibited naïve T cell differentiation into effector T cells (Fig. 3h, i). Importantly, rSOD3 significantly reduced the

proportion of IFN-γ-producing T cells in vitro, suggesting that SOD3 directly inhibited the expansion of IFN-γ-producing T cells (Fig. 3j).

To further elucidate the molecular mechanism of T cell inhibition by SOD3, we prescreened and compared the expression of the sixteen proteins that regulate IFN-γ signaling in splenic cells in SOD3⁻/⁻ and WT mice (Supplementary Fig. S4a). Elevated expression of nuclear factor kappa B (NF-κB), the signal transducer and activator of transcription 5 A (STAT5a) and c-Jun N-terminal Kinase (JNK) was identified in spleen from SOD3⁻/⁻ mice compare to WT mice after infection, whereas reduction of NAD-dependent deacetylase sirtuin-1 (SIRT1) and mitogen-activated protein (MAP) kinases/P38 expression was observed in SOD3⁻/⁻ mice compare to WT mice. The expression of five toll-like receptors, STAT5b and protein kinase B (PKB, or Akt) was relatively stable in the two experimental groups (Supplementary Fig. S5a). Importantly, an increased expression of JNK was observed in purified T cells in SOD3⁻/⁻ mice after infection (Fig. 3k, l), and JNK has been known to control activation of NF-κb and STAT5a, we thus focused on its involvement in the regulation of IFN-γ expressing T cells. Inhibition JNK activity with a specific inhibitor SP600125 impaired

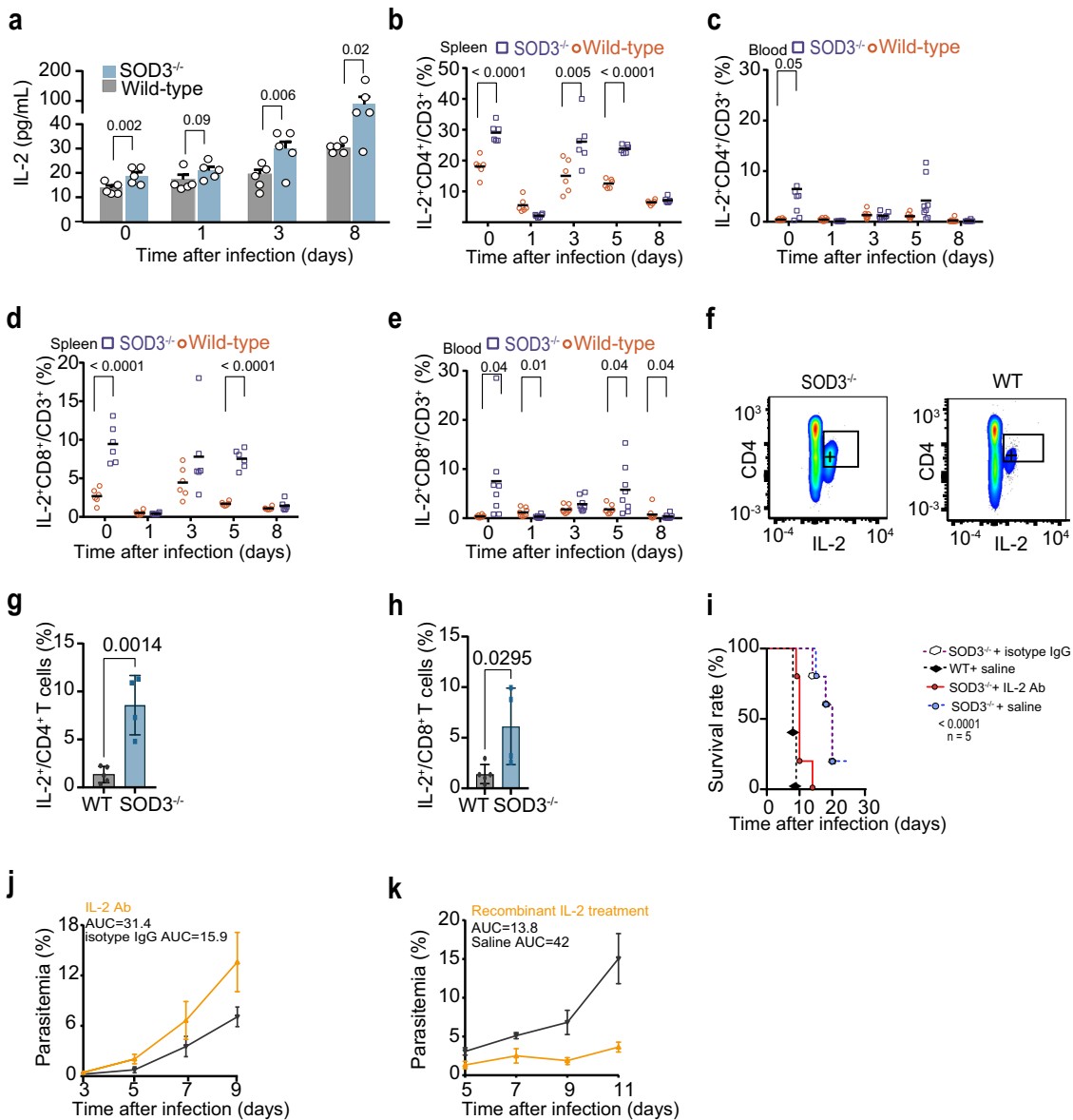

**Fig. 4 | SOD3 suppresses IL-2 expression in T cells and immune resistance in *Plasmodium* infection. a** IL-2 levels in mouse sera were gradually elevated in SOD3$^{-/-}$ compared with WT mice after infection by *P. berghei* ANKA (*n* = 5). **b–e** Statistical analysis of IL-2 production in splenic and circulating CD4$^+$ and CD8$^+$ cells from both WT and SOD3$^{-/-}$ mice (*n* = 6–8) at different time points after *P. berghei* infection. **f** Representative flow cytometry plots of splenic IL-2$^+$ CD4$^+$ T cells in both SOD3$^{-/-}$ mice and WT mice. **g, h** Statistical analysis of IL-2 production in CD4$^+$ and CD8$^+$ T cells from both WT (*n* = 5) and SOD3$^{-/-}$ mice (*n* = 4) after *P. y. yoelii* infection. **i, j** Neutralization of IL-2 in SOD3$^{-/-}$ mice resulted in early death and

higher parasitemia after *P. berghei* infection compared to control mice. Five biological replicates were included in the analysis. Parasitemia was depicted with the area under curve (AUC). The survived mice in SOD3$^{-/-}$ +saline group and SOD3$^{-/-}$ +IgG were humanely euthanized accordingly to approved criteria. **k** Infusion of recombinant IL-2 in WT mice inhibited *P. berghei* proliferation (*n* = 5). Parasitemia was depicted with the AUC. Histograms present the mean ± SD. For (**a, g, h**), Kaplan-Meier survival curve p-values were performed using Log rank Mantel-COX test. Source data are provided as a Source Data file.

proliferation of IFN-γ-producing T cells (Fig. 3m). Taking together, SOD3 was found to inhibit the expression of JNK in T cells and JNK activation is crucial for IFN-γ-mediated T-cell toxicity.

## SOD3 suppressed IL-2 expression by T cells

IL-2 is a major T cell growth factor which promotes the expansion of IFN-γ expressing T cells[19]. Here, we observed a time-dependent increase in the serum IL-2 in SOD3$^{-/-}$ mice after parasite infection, but not IL-10, TNF-α, compared to that of WT mice (Fig. 4a, Supplementary Fig. S5a, b). Further, higher percentages of IL-2$^+$ CD4$^+$ T cells and IL-2$^+$ CD8$^+$ T cells in SOD3$^{-/-}$ mice than in WT mice persisted until late infection (Fig. 4b–e). Moreover, an increased proportion of IL-2-producing CD4$^+$ T cells and CD8$^+$ T cells was also observed in SOD3$^{-/-}$

mice after *P. y. yoelii* infection (Fig. 4f–h). Importantly, IL-2 depletion by infusion of IL-2-specific antibodies in vivo significantly increases susceptibility to *Plasmodium* infection (Fig. 4i, j). And intravenous injection of recombinant IL-2 rescued resistance to infection by significantly reduced parasitemia in WT mice (Fig. 4k), confirming that IL-2 plays a protective role in resistance to *Plasmodium* infection in SOD3$^{-/-}$ mice.

As SOD3 binds CD4$^+$ T cells and CD8$^+$ T cells (Fig. 3a), we therefore tested whether SOD3 inhibits T cells to generate IL-2. Addition of SOD3 into the splenic lymphocytes significantly reduced the proportion of IL-2-producing CD4$^+$ T cells and CD8$^+$ T cells, suggesting that SOD3 can directly regulate the expansion of IL-2-expressing T cells (Fig. 5a, b). To investigate whether the inhibition of

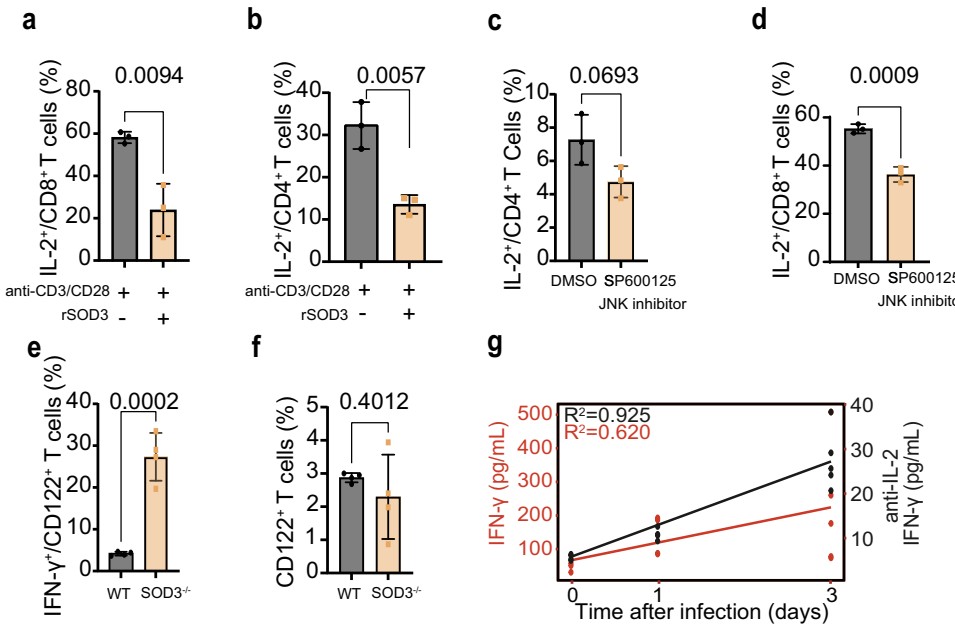

**Fig. 5 | SOD3 suppresses IL-2 and concomitant IFN-γ production by T cells.**
**a**, **b** The proportion of CD8⁺ or CD4⁺ T cells producing IL-2 upon treatment with
SOD3 was significantly reduced in in vitro experiments, $n = 3$. **c**, **d** Treatment with
SP600125, a selective JNK inhibitor, led to a decrease in the proportion of IL-2-
producing CD8⁺ or CD4⁺ T cells, $n = 3$. **e** and **f** The proportion of IFN-γ⁺CD122⁺ T cells

but not that of CD122⁺ T cells expanded in the SOD3⁻/⁻ mice, $n = 4$ (**e**, **f**). **a–f** Two-
tailed student's $t$ test was used to test for significant differences between two
groups. **g** Serum levels of IFN-γ was analysis by neutralizing Abs for IL-2, $n = 3–4$.
Histograms present the mean ± SD. Source data are provided as a Source Data file.

IL-2-expressing T cells by SOD3 is causally related to JNK activation,
we assessed the proportion of IL-2-producing T cells under JNK-
deficient conditions. Inhibition of JNK activity led to the reduction of
IL-2-producing CD4⁺ and CD8⁺ T cells (Fig. 5c, d). These data indicate
that SOD3 can directly suppress the expansion of IL-2-producing
CD4⁺ and CD8⁺ T cells, which may require JNK activation. SOD3
deficiency significantly increased the proportion of IFN-γ expressing
CD122⁺ (IL-2 receptor) T cells without altering the proportion of
CD122⁺ T cells (Fig. 5e, f). Furthermore, neutralization with IL-2-
specific antibody resulted in a striking reduction of IFN-γ levels in the
sera (Fig. 5g). Overall, our data suggest that SOD3 suppresses T cells
by inhibition of IL-2 autocrine pathways, thereby benefits parasites in
evading host immune surveillance.

**Neutrophils are the main cellular sources of SOD3**
To identify the source of splenic SOD3, we analyzed available single-
cell transcriptome data (Mouse Cell Atlas, https://bis.zju.edu.cn/MCA/)
from healthy mice and the data indicated that SOD3 was primarily
expressed in splenic macrophages and neutrophils. Correspondingly,
we found, in WT mice, the proportions of CD11bʰⁱ macrophages
(Fig. 6a) and neutrophils (Fig. 6b) after infection doubled compared to
uninfected controls, while other subpopulations decreased (Fig. 6c, d).
We thus sorted splenic macrophages and neutrophils at different time
points post-infection using fluorescence-activated cell sorting and
examined the expression of SOD3 in these cells. The results of
RT−qPCR, immunofluorescence and Western blotting illustrated that
only neutrophils showed a significant increase in SOD3 mRNA tran-
scription (Fig. 6e, f) and protein expression after infection (Fig. 6g, h).
In agreement with the above results, the adoptive transfer of neu-
trophils, but not macrophages, from WT mice into SOD3⁻/⁻ mice sig-
nificantly reduced the survival time of SOD3⁻/⁻ mice after parasite
infection (Fig. 6i, j). These data strongly suggest that the parasites
evaded immune surveillance by inducing SOD3 secretion from neu-
trophils, which suppress IL-2 and IFN-γ-mediated host immune
responses to protozoa infection.

## Discussion
Pathogen infection triggers oxidative stress by generating an imbal-
ance between the oxidant and antioxidant systems. SOD3, a member
of the superoxide dismutase family, has been primarily recognized as
an enzyme for scavenging ROS in extracellular spaces. In this study, we
revealed a function of SOD3, which was upregulated during protozoan
infection. However, the overexpression of SOD3 was surprisingly
found to dysregulate the host innate immune response to invading
parasites. SOD3, primarily expressed by neutrophils, can directly bind
and suppress the expansion of IL-2-expressing T cells, concomitantly
reducing the expansion of IFN-γ-expressing T cells, possibly by inhi-
biting the JNK pathway. This eventually results in significantly wea-
kened host immune resistance to parasite infection (Figs. 1–6,
Supplementary Fig. S6). Overall, our results revealed an unexpected
link between invading parasites and host neutrophils, highlighting that
SOD3 could be a target for mitigation of inflammation and hyperten-
sion in the context of infectious diseases.

An essential pathophysiological role of SOD3 has been noticed in
inflammatory diseases and autoimmune disorders[20–22]. However, there
have been no systemic investigations on the relevance between SOD3
and parasite infection. Here, we systematically scrutinized the invol-
vement of SOD3 in the suppression of early innate immune reaction
after parasite infection and deciphered the underlying mechanism.
The data suggest that SOD3 acted as an immune dysregulatory factor
in the infections, which is supported by the fact that SOD3⁻/⁻ mice were
significantly more resistant than their WT littermates to infection with
either lethal or nonlethal parasite species (Figs. 1 and 2). IL-2 derived
from CD4⁺ T cell seemed to play a critical role in the SOD3⁻/⁻ mice, as
neutralization of IL-2 with specific antibodies significantly reduced the
survival time of the mice compared with the controls after parasite
infection. Furthermore, the fact that the preferential binding of SOD3
to T cells and inhibition of the expansion of IL-2 expressing or IFN-γ-
producing T cells and downstream IFN-γ production also strongly
indicated SOD3 played an important regulation in immune responses
to protozoa infection.

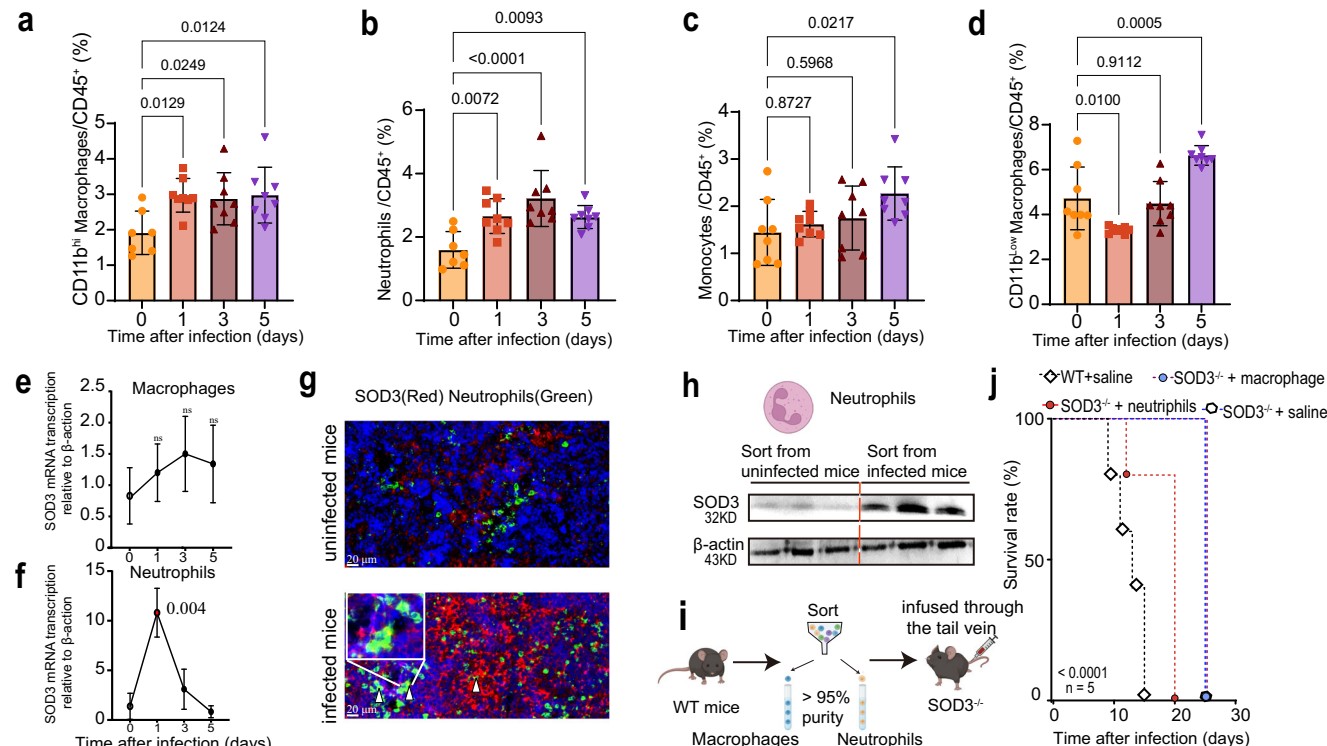

**Fig. 6 | Neutrophils are the main cell sources of SOD3 in response to parasite infection. a–d** Statistical analysis of CD11b$^{hi}$F4/80$^+$ monocyte-derived macrophages (**a**), neutrophils (**b**), monocytes (**c**), and CD11b$^{low}$F4/80$^+$ macrophages (**d**) from WT mice at different time points after *P. berghei* ANKA infection. n = 7-8 (a-d) Multiple comparisons test corrected for Tukey was used to test for significant differences between uninfected and infected groups. **e, f** Transcription of the gene coding for SOD3 was determined by RT−qPCR in neutrophils (**e**) and macrophages (**f**) after parasite infection (*n* = 3). An ~10-fold increase in gene transcription was observed in neutrophils but not in macrophages. Statistical significance tested by one-way ANOVA, and multiple comparisons to the uninfected control were corrected using Dunnet's method. Three biological replicates were included in the experiment. **g** Representative high-magnification images of spleen sections of WT mice analyzed with double immunofluorescence for SOD3 (red) and myeloperoxidase (MPO, green). Significant augment of SOD3 expression in the splenic tissue of infected mice was observed. Arrow heads indicate representative cells with strong colocalization. Three biological replicates were included in the experiment. Scale bar =20 μm. **h** Comparable Western blotting analysis of SOD3 expression in neutrophils sorted from uninfected or infected mice. SOD3 expression in neutrophils of infected mice was significantly more than that from uninfected mice. Three biological replicates were included in the experiment. Each experiment was repeated three times independently with similar results. The neutrophils illustration was created with http://BioRender.com (publishing license: TB26UIITUU). **i** Schematic representation of the procedure for adoptive transfer. The illustration was created with http://BioRender.com (publishing license: NC26UIHFCR). **j** Adoptive transfer of WT neutrophils into SOD3$^{-/-}$ mice resulted in early death of SOD3$^{-/-}$ mice after *P. berghei* ANKA infection. Five biological replicates were included in the analysis. Kaplan–Meier survival curves were calculated using the survival time for each mouse in all groups, and significance was determined by the log-rank test. Histograms present the mean ± SD. Source data are provided as a Source Data file.

Studies have strongly suggested that an early burst of IFN-γ production is associated with protection to *P. falciparum* and *P. berghei* ANKA infection[23,24]. Parasite-related HIFs have been reported to facilitate immune evasion by inhibiting IFN-γ-mediated pathogen clearance[3,25–28]. Here, we showed that IFN-γ responses in SOD3$^{-/-}$ mice were augmented and neutralization of IFN-γ in SOD3$^{-/-}$ mice at early infection resulted in reduced survival time in 80% of SOD3$^{-/-}$ mice. Moreover, neutralization of IL-2 at early time also led to decreased IFN-γ production in SOD3$^{-/-}$ mice in response to parasite infection. Thus, the protective effect of SOD3 deficiency is mediated by early production of IFN-γ. The observations in this study demonstrate important differences in baseline immunity between SOD3 null mice and their WT control, suggesting that SOD3 efficiently targets IFN-γ and IL-2 producing T cells. These data support the conclusion that SOD3 is a pivotal immune factor in modulation of host innate immune responses to parasite infection by controlling the production of IL-2 and IFN-γ.

While we believe that these results presented in this study is an advance in understanding the interaction between the invade protozoan parasites and host immune system, the exact mechanism underlies the possible regulation of SOD3 on the T cells remains further investigation. The reason is that, even though the active SOD3 protein showed direct inhibition on T cell function in both in vitro and in vivo conditions, other functions of SOD3 may mediate the effects independent of its enzymatic activity and affinity to the immune cells. Further, a significant increase in JNK expression was indeed observed in T cells sorted from SOD3$^{-/-}$ mice after infection, there may be an unknown link between SOD3 and the kinase activity of the JNK protein.

Overall, this study expands our understandings of how neutrophil-derived SOD3 can modulate immune responses and argue for the importance in keeping the balance between the antioxidant activities and the potent immune responses to invading pathogens.

## Methods

### Ethics
The animal experiments were conducted according to the animal husbandry guidelines of Shenyang Agricultural University (permit no. SYXK<Liao>2021-0010). Human sera samples were obtained under approved protocols of the Ethical Committee of the Chinese Academy of Medical Sciences (approval no. IPB-2016-2) and Institutional Review Board (no. PRAMS0034319) of the Pennsylvania State University.

### Animals
Female and male C57BL/6 mice were obtained from Liaoning Changsheng Biological Technology Company (Liaoning, China). C57BL/6-

Sod3tm1cyagen (SOD3$^{-/-}$) mouse strains (Serial Number: KOCMP-22050-Sod3) were purchased from Cyagen Biosciences (Suzhou, China). Mice were maintained under the following housing conditions: ambient temperature 22 °C, humidity control 50%, 12 h light/12 h dark cycle. Littermates used in the tests were of the same sex and similar body weights as the SOD3$^{-/-}$ mice. SOD3 transgenic mice (Global SOD3 overexpression) were constructed with the help from GRENSTER *Co., Ltd.* (Liaoning, China) and achieved by breeding a CAG-SOD3-IRES-ZsGreen1 mouse to transgenic mice, which globally overexpressed SOD3, with an average protein expression level -1.4-fold higher than normal in serum. In this study, we have used humane endpoints for the infected animals. To determine when the animals should be euthanized, we used the specific signs such as weight loss, inability to rise or ambulate, or dehydration. In Kaplan-Meier survival analysis, mice that are euthanized in humane endpoints are considered "censored" data to distinguish them from mice where death actually occurred.

### Sera from *P. falciparum* malaria patients and healthy donors

All procedures performed on human samples were carried out in accordance with the tenets of the World Medical Association's Declaration of Helsinki. A total of 20 patients (ten male and ten female) suffering from *falciparum* malaria were recruited from a malaria-endemic area, all patients provided with written informed consent as previously described[28]. Another serum samples were collected from *P. falciparum*–infected patients (24 male)[29] with a recent history of travel to malaria-endemic African countries, and *P. falciparum* infection was clinically confirmed by a microscopy. The inclusion criteria were no history of *P. falciparum* infection. Healthy control sera were collected from healthy volunteers with informed consent. The information of all individuals involved was anonymized. Written consent for the publication of this study was obtained from all individuals.

### Experimental malaria models

*Plasmodium* species lethal to rodents, including *P. y. yoelii* YM, *P. y. yoelii* 17XL and *P. berghei* ANKA, were donated by Dr. Yaming Cao and propagated by passage in C57BL/6 male mice of 7−8 weeks old; a transgenic *P. berghei* ANKA parasite strain expressing luciferase under the control of the *ef1a* promoter (*Pb*A luc)[28] was provided by Dr. Wenyue Xu (Army Medical University, China) and propagated by passage in mice.

Preliminary ECM experiments in mice were performed to optimise the infection dose. Of the four doses tested ($10^2$, $10^3$, $10^4$, $10^5$ iRBCs), the minimum infection dose of $10^3$ *P. berghei* ANKA parasites was collected and injected intraperitoneally into 7−8-week-old male C57BL/6 naïve mice to initiate ECM experiments, minimizing pain and distress to the mice. Humane endpoints were defined as serious neurological signs or immobility. Parasitemia was determined by examination of Giemsa-stained thin blood smears by light microscopy every other day after infection.

### Detection of the accumulation of malaria parasites in the mouse brains by in vivo imaging

Upon *P. berghei*–luc infection, the accumulation of parasites in the brain and in the body was assessed in SOD3$^{o/e}$ mice as well as SOD3$^{-/-}$ mice and WT mice with an AniView600 multimode in vivo animal imaging system (Guangzhou Biolight Biotechnology *Co., Ltd.*, China) after intraperitoneal injection of D-Luciferin (High Purity, Biyuntian, China, Cat# ST198). At least three mice were used in each group.

### *T. brucei* cultivation and infection

The *T. brucei* Lister 427 strain[30] was donated by Dr. Zhaorong Lun (Sun Yat-sen University, China) and maintained in our laboratory. The bloodstream form of *T. brucei* was proliferated in Hirumi's modified Iscove's-9 medium and propagated by passage in mice. SOD3$^{-/-}$ mice and WT mice were inoculated intraperitoneally with 50 *T. brucei*

parasites. Parasitemia was determined by evaluating tail blood by light microscopy with a hemocytometer each day after infection.

### *T. gondii* RH cultivation and infection

The *T. gondii* RH strain was maintained in our laboratory. The parasites were expanded in *Vero* cells and propagated by passage in mice. *T. gondii* RH strain tachyzoites were grown under the indicated conditions and collected by filtration as previously reported[31]. SOD3$^{-/-}$ mice and WT mice were inoculated intraperitoneally with $1 \times 10^4$ tachyzoites.

### In vivo IFN-γ and IL-2 depletion

For in vivo depletion of IFN-γ and IL-2, SOD3$^{-/-}$ mice were injected through the tail vein with anti−IFN-γ monoclonal antibody (mAb) (Clone R4-6A2, Selleck Chemicals, USA) and anti−IL-2 mAb (Clone JES6-1A12, Leinco Technologies, St. Louis, MO, every other day). SOD3$^{-/-}$ mice in the control group were injected intraperitoneally with the corresponding isotype control IgG. Parasitemia was determined by examination of Giemsa-stained thin blood smears by light microscopy every other day after infection. The parasitemia was quitted after the total death of the animals in the control group.

### SOD3 rescue experiment

For rescue experiments, SOD3$^{-/-}$ mice were injected with recombinant mouse active SOD3 protein every other day from day 0–10 after *P. berghei* infection (Cat# APA117Mu01, Cloud clone crop, China, purity> 95%). The injected concentrations of SOD3 corresponded to that of SOD3 in WT mice after infection. SOD3 has a long half-life of - 20 h, and the endotoxin level was determined by the limulus amoebocyte lysate method. The specific activity of recombinant mouse SOD3 is 273.8 U/mg. SOD3$^{-/-}$ mice in the control group were administered with the same volume of saline buffer via tail vein injection.

### Adoptive transfer of neutrophils and macrophages from WT mice to SOD3$^{-/-}$ mice

Splenic neutrophils and macrophages were positively selected as before[32]. The purity of sorted cells, as verified by FACS analysis, was more than 95%, which separated intact live cells from dead cells and enucleated cellular debris. The proportion of dead cells was determined by using a hemocytometer to count the cells stained with trypan blue and was lower than 5%. Fifteen SOD3$^{-/-}$ mice were randomly allocated to three groups with five animals in each group. In the SOD3$^{-/-}$ + WT macrophage group, SOD3$^{-/-}$ mice (recipients) received $1.5 \times 10^6$ macrophages from WT mice *i.v.* (SOD3$^+$ macrophages→SOD3$^{-/-}$ host). In the SOD3$^{-/-}$ mice + WT neutrophils group, SOD3$^{-/-}$ mice (recipients) received only $1.5 \times 10^6$ neutrophils from WT mice *i.v.* (SOD3$^+$ neutrophils→SOD3$^{-/-}$ host). SOD3$^{-/-}$ mice and WT mice in the control group received the same volume of saline buffer. Meanwhile, mice in all groups were inoculated intraperitoneally with iRBCs.

### Immunofluorescence and immunohistochemical analysis of SOD3 expression in splenic and brain tissues of WT and SOD3$^{-/-}$ mice

For immunofluorescence analysis, mice were sacrificed at day seven after *P. y. yoelii* YM infection, and the spleens were fixed in 4% paraformaldehyde, embedded in paraffin, and sectioned. In brief, spleen sections were blocked with 10% FBS in PBS, incubated with anti-SOD3 antibodies (Affinity Biosciences, China, Cat# DF7753) and anti-myeloperoxidase primary antibodies (MPO, Abcam, UK, Cat# ab208670) in 5% FBS in PBS for 1 h at 37 °C, washed five times with PBS, and incubated with the corresponding secondary antibodies according to the IF staining procedure. The same procedure was performed to stain healthy spleen sections.

For immunohistochemical analysis, *P. berghei* ANKA-infected mice were sacrificed and the brain and spleen were fixed in 4% paraformaldehyde, embedded in paraffin, and sectioned. Brain sections

were blocked with 10% FBS in PBS, incubated with anti-SOD3 antibodies (Affinity Biosciences, Cat# DF7753, RRID: AB2841219), incubated with a corresponding secondary antibody, and then stained with 3,3-diaminobenzidine. Spleen sections were incubated with a rabbit anti-IFN gamma antibody (Bioss, Beijing, China, Cat# bs-0480R). The same procedure was performed to stain healthy brain and spleen sections.

## Detection of SOD3 in the sera of both parasite-infected mice and humans using enzyme-linked immunosorbent assay (ELISA)

The concentrations of SOD3 in the sera of WT mice at different time points post-infection were detected using a mouse SOD3 ELISA kit (CLOUD-CLONE CORP., Cat# SEA117Mu) according to the manufacturer's instructions. Serum samples require about a 100-fold dilution. Three healthy mice were used as controls. Briefly, 100 μL of different concentrations of standards and samples were added to each well. The wells were then covered with an adhesive strip and incubated for 2 hours at 37 °C. After removing the liquid from each well, 100 μL of biotin antibody (1x) was added to each well after three washes with PBS. The wells were covered with a adhesive strip and incubated for 1 hour at 37 °C. Next, each well was aspirated and washed three times. After washing, 100 μL of HRP-avidin (1x) was added to each well and incubated for 1 hour at 37 °C, followed by five washes. Finally, after the addition of TMB substrate, the optical density (OD) at 450 nm was measured. The minimum detectable dose of SOD3 is typically less than 1.27 ng/mL.

The concentrations of SOD3 in the sera of healthy volunteers and *P. falciparum*–infected patients were detected using a Human SOD3 ELISA kit (Elabscience, Wuhan, China, Cat# E-EL-H2382) according to the manufacturer's instructions. Same experimental process was followed as explained above.

## LC–MS/MS analysis of serum SOD3

Serum proteins were extracted from WT and SOD3$^{-/-}$ mice, and the high-abundance proteins in the sera were removed using the ProteoMiner protein enrichment kit (Bio-Rad, USA), followed by enzymatic digestion, enrichment of informative peptides, and LC–MS/MS (Orbitrap Exploris 480) for separation and detection. The search parameters were set as follows: mass tolerance for precursor ion was 10 ppm, and mass tolerance for product ion was 0.02 Da. SOD3 protein levels were calculated using Proteome Discoverer 2.4 (Thermo). The identified SOD3 protein contains at least 1 unique peptide.

## Statistics and reproducibility

Student's *t*-test was used to calculate significance between two groups and ANOVA test was used to calculate differences for more than two groups. All data are shown as mean values and errors bars represent standard deviation. Each replicate was biologically independent. The sample size and number of replicates were determined based on previous studies.

Pearson correlation and regression analyses were used for the correlation studies. Correlation analysis was performed using the function "cor" of the base R package by applying Pearson's method, using default parameters unless otherwise specified. R (version 4.0.3) and Graph Pad Prism (version 9) was used for all statistical analyses. Uncropped blots are found in Supplementary Fig. 7. The gating strategy is given in Supplementary Fig. 8.

## Reporting summary

Further information on research design is available in the Nature Portfolio Reporting Summary linked to this article.

## Data availability

All data supporting the findings of this study are available within the paper. Source data are provided with this paper.

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

## Acknowledgements

This research was supported by a grant from the National Key Research and Development Program of China (grant number 2022YFD1800200) to Q.C., the National Nature and Science Foundation of China (grant number 82030060) to Q.C., the CAMS Innovation Fund for Medical Sciences (CIFMS) (grant number 2019-I2M-5-042) to N.J., and the NIH grant (U19 AI089672) to L.C. Figures 1a, 6h, i, Figures S2d, and S6 Created with BioRender.com released under a Creative Commons Attribution-NonCommercial-NoDerivs 4.0 International license.

## Author contributions

Q.L. performed most experiments, analyzed the data, and wrote the first draft of the manuscript. N.J. supervised the study. K.L. performed all Western blot and assisted with animal experiments, T.L. performed in vitro experiments using primary T cell, Y.Y., L.Y., Z.S., Y.Z., and A.F. assisted with the animal experiments. N.H., L.C., and Y.C. were responsible for collection of serum and confirmation of infection. W.X provided the luc-expression *P. berghei* parasite strain. L.Y., Z.S., N.H., and A.F. performed the infection experiment. X.S., Y.F., and R.C. assisted with the flow cytometry experiments. Q.C. conceived the study, analyzed the data, and finalized the manuscript.

## Competing interests

The authors declare no competing interests.
