## [Peer Review File · Nature Communications]

SOD3 suppresses early cellular immune responses to parasite infectionReviewers' comments:

Reviewer #1 (Remarks to the Author):

This manuscript explores the role of superoxide dismutase 3 (SOD3) in mediating disease pathogenesis in several mouse models of protozoan infection, including trypanosomiasis, toxoplasmosis and malaria, with extensive application of the virulent malaria model of cerebral malaria and hyperparasitemia, that uses *Plasmodium berghei* ANKA. Using SOD3^{-/-} (null) mice, the authors assess infection outcomes and a number of immune parameters, comparing intact (WT) and null mice. Adoptive transfer, antibody ablation and SOD3 overexpression are also applied. While the idea and some of the data are quite compelling, and convincingly show evidence that SOD3 plays a significant role in determining outcomes of infection in the various models, the immunological data are very difficult to interpret as shown, statistical analysis is in some cases poorly applied and in many cases difficult to follow, figures are inadequately described, and the results are over- or inaccurately interpreted. Thus, the mechanism of action of SOD3 is difficult to ascertain.

Materials and methods

The ethics statement appears to be incomplete.

The experimental cerebral malaria (ECM) model is poorly described. How the development of this syndrome was tracked and monitored, or what indicators were used to assess it, and what humane endpoints were used are not described. Death as an endpoint in this model does not meet humane euthanasia criteria.

Data demonstrating the overexpression of SOD3 in the mouse model should be shown.

The injection of 1×10^3 parasites intraperitoneally in the various malaria models is quite low relative to most published work. The authors should discuss and justify this.

In adoptive transfer experiments, did the authors ensure that the neutrophils were not activated before delivery to recipient mice? How so?

Correlation analyses and regression analysis are presented in the Results but are not described in Methods. This is important because the application of these tests is poorly described otherwise. In general, statistical analyses are very difficult to interpret in this manuscript and text descriptions of the data are difficult to follow. Parasitemia curves should not be compared day by day with t tests, as data are not independent and have a time component; the authors should consider area under the curve in these cases. A statistician should be consulted to review all of the data and analyses in this paper.

Results

The survival curves shown in this work are the most exciting aspect of the study and are compelling.

In all cases where mice were immunologically manipulated (ablation experiments, for example), impacts on parasitemia should be shown and considered. This was done only in Fig 4 (IL-2 manipulation), albeit without showing a complete course of infection (graphs stop at days 9 and 11, yet some mice survived to day ~36)..

None of the western blots shown are full blots, but are rather excessively trimmed. Full blots should be offered as supplemental data.

Throughout the Results section, immunological and other biochemical measurements are done very early in infection at time points that do not correspond to when mice succumb to infection. This makes it difficult to attribute the described changes to the observed outcomes. Examples include Fig 1i, Fig 2, Fig 3, Fig 4, and some of Fig 5.

Fig 1 does not make a compelling case that SOD3 is a strong determinant in outcomes in toxoplasmosis.

Specific criticisms of data analysis: Fig 1b, t test is not appropriate for multiple comparisons against day 0; Fig 2g also needs correction for multiple hypothesis testing within the data; Fig 2a, Fig 3a and Fig 4l results are impossible to interpret as presented and described; Fig 1e should be presented as a full course of infection with comparison of area under the curve; sample sizes are not consistently provided in figure legends and time points at which data were collected are not provided (including but not limited to Fig 2f, Supp Fig 2, Supp Fig 3, etc).

Other problems with graphs: Fig 2b and c don't seem to be representing the same data (ie changes in expression level of SOD3 are unconvincing in the blot); densitometry for Fig 3h should be provided, and the interpretation of such data reviewed as the text does not match what is shown in the graph, especially for TLR7 and TLR8; Fig 4c-f need to be labeled to identify source of cells;

Fig 5a-d need a legend to identify which groups are being shown (and it is difficult to match what is described in the text to what is shown in these panels); Supp figures are misnumbered in the reviewer interface; Supp Fig1f, spleen index and how it was calculated is not described anywhere; histological features claimed to be evident in Supp Fig 2 are not evident – the images are low power and no arrows or other identifying marks are used; Supp Fig 3 is poorly described and arrowheads are not defined in the legend; the text description of Supp Fig 4 does not seem to correspond to the data shown in the actual figure.

Data interpretation and discussion

Throughout this manuscript, various timelines are chosen for representation in the figures without justification. For the most part, the immunologic changes shown are from early timepoints in the infections, yet survival curves show very extended times to death, in some cases weeks later. Thus, it becomes difficult to associate the immune changes with disease outcomes. While it is appreciated that WT mice succumb quickly to ECM and the authors attempted to compare events in those mice with those in null mice, the latter DO eventually succumb to infection (in all models shown except adoptive transfer in Supp Fig 3), and there is no examination whatsoever in the manuscript to explore what ultimately does induce death for these mice. Another important issue is that the null mice, by most criteria measured, are at baseline immunologically quite different than their WT counterparts. The authors are far as this reviewer can tell, never discuss this point clearly, nor consider the implications.

The authors make some claims about causation and temporal relationships that are either not born out by the data or are not evident in the data as shown, for example lines 253-5, 288-9, 361-3.

Reviewer #2 (Remarks to the Author):

Authors examined the role of SOD3, extracellular ROS scavenger, in Plasmodium infection and other protozoan infections. Authors showed that SOD3-deficient mice are resistant to Plasmodium infection, while SOD3-overexpressing mice are more vulnerable. Authors suggest that SOD3 regulate cytokine production by host immune cells and regulate host immunity during protozoan infection. Authors described experiments using multiple species of protozoan parasites in mice and in human malaria infection. However, each experiment is poorly described and lacks the critical information that justify their conclusion. I advise authors to focus this study on malaria alone and describe details of the methods.

In addition, authors did not address how SOD3 activity is linked to the inhibition of cytokine production by T cells and NK cells, while they showed that cytokine production was improved in SOD3 KO mice in malaria infection model. Authors only mentioned that SOD3 could directly repress the activation of T cells (line 355).

More specifically,

1. Introduction lacks background information of the study. What is known about SOD3 and what has been investigated? Why understanding of redox signaling that facilitate pathogen evasion of host immune response is challenging? What are known regarding SOD1 and SOD2?
2. References of the experimental animals used in this study should be indicated.
3. Information of human studies should be described in much more in detail. How were malaria patients recruited? What was the criteria of selecting 20 patients? What are the names of the hospitals? What are age and sex of these patients and healthy volunteers?
4. Information of parasites used in the study should be described in much more detail and references should be indicated. Write full name of each parasite and how authors obtained these materials.
5. The amount of monoclonal antibody used for in vivo is too low for neutralization.
6. Methods used for SOD3 detection should be explained much more in detail.

7. Parasitemia should be determined throughout the course of the infection with parasites, not just one time-point.
8. Approximately 50% of SOD3 KO mice died in 14 days after *P. berghei* infection. What was the parasitemia levels? Why did these mice died without parasite accumulation in the brain (Fig. 2d)?
9. For all flow cytometry analysis, show gating strategy and representative flow cytometry profiles.
10. Authors should also examine Plasmodium antigen-specific cytokine production by T cells.
11. Show reference of single-cell transcriptome data of SOD3 expression (line 290-292).
12. Explain the experiment in Fig. 5i in more detail. Did authors prepare macrophages and neutrophils from uninfected mice? Are they from spleen? Did they transfer to infected mice? What day after infection? What was the purity?

Answers to Questions and Comments from the reviewers:

Reviewer #1:

Comment 1: *This manuscript explores the role of superoxide dismutase 3 (SOD3) in mediating disease pathogenesis in several mouse models of protozoan infection, including trypanosomiasis, toxoplasmosis and malaria, with extensive application of the virulent malaria model of cerebral malaria and hyperparasitemia, that uses Plasmodium berghei ANKA. Using SOD3^{-/-} (null) mice, the authors assess infection outcomes and a number of immune parameters, comparing intact (WT) and null mice. Adoptive transfer, antibody ablation and SOD3 overexpression are also applied. While the idea and some of the data are quite compelling, and convincingly show evidence that SOD3 plays a significant role in determining outcomes of infection in the various models, the immunological data are very difficult to interpret as shown, statistical analysis is in some cases poorly applied and in many cases difficult to follow, figures are inadequately described, and the results are over- or inaccurately interpreted. Thus, the mechanism of action of SOD3 is difficult to ascertain.*

Answer to comment 1: We appreciate very much for the positive comment and the manuscript has been revised extensively with more experiments supplemented.

Comment 2: *The ethics statement appears to be incomplete.*

Answer to comment 2: We apologize for this confusion. Due to the requirement of the “Double-blind peer review”, we deleted the detail of the ethics statement in the previous submission. We have now added a full ethics statement to this manuscript.

“The animal experiments were conducted according to the animal husbandry guidelines of Shenyang Agricultural University (permit no. SYXK<Liao>2021-0010). Studies in humans were reviewed and approved by the Ethical Committee of the Chinese Academy of Medical Sciences (approval no. IPB-2016-2) and Institutional Review Board (no. PRAMS0034319) of the Pennsylvania State University.”

Comment 3: *The experimental cerebral malaria (ECM) model is poorly described. How the development of this syndrome was tracked and monitored, or what indicators were used to assess it, and what humane endpoints were used are not described. Death as an endpoint in this model does not meet humane euthanasia criteria.*

Answer to comment 3: We are sorry for the unclearness. We have revised the description of experimental cerebral malaria model.

“In this study, we have used humane endpoints for the infected animals. To determine when the animals should be euthanized, we used the specific signs such as weight loss, inability to rise or ambulate, or dehydration.”

Method: “For ECM model, the humane endpoints were defined as serious neurological signs or immobility.”

Comment 4: *Data demonstrating the overexpression of SOD3 in the mouse model should be shown.*

Answer to comment 4: We are sorry for the unclearness. The data was shown in both Method and Result sections.

Comment 5: *The injection of 1 x 10³ parasites intraperitoneally in the various malaria models is quite low relative to most published work. The authors should discuss and justify this.*

Answer to comment 5: We are sorry for the unclearness. The reasons for the selection are as follows: 1) The injection of 1 x 10³ parasites intraperitoneally was indeed conducted based on previous publications¹⁻³. 2) The minimum infection dose of 10³ P.

berghei ANKA parasites was selected for minimizing disturbance and distress to the mice. 3) Other malaria models were consistent with the minimum infection dose of 10^3 *P. berghei* ANKA.

“Preliminary ECM experiments in mice were performed to optimise the infection dose. Of the four doses tested (10^2 , 10^3 , 10^4 , 10^5 iRBCs), the minimum infection dose of 10^3 *P. berghei* ANKA parasites was collected and injected intraperitoneally into 7–8-week-old male C57BL/6 naïve mice to initiate ECM experiments, minimizing pain and distress to the mice.”

References:

1. Nolasco-Pérez TJ, Cervantes-Candelas LA, Buendía-González FO, Aguilar-Castro J, Fernández-Rivera O, Salazar-Castañón VH and Legorreta-Herrera M (2023) Immunomodulatory effects of testosterone and letrozole during *Plasmodium berghei* ANKA infection. *Front. Cell. Infect. Microbiol.* 13:1146356. doi: 10.3389/fcimb.2023.1146356
2. Zafar I, Taniguchi T, Baghdadi HB, Kondoh D, Rizk MA, Galon EM, Ji S, El-Sayed SAE-S, Do T, Li H, Amer MM, Zhuwei M, Yihong M, Zhou J, Inoue N and Xuan X (2023) Babesia microti alleviates disease manifestations caused by *Plasmodium berghei* ANKA in murine co-infection model of complicated malaria. *Front. Cell. Infect. Microbiol.* 13:1226088. doi: 10.3389/fcimb.2023.1226088
3. DING Y, XU W, ZHOU T, LIU T, ZHENG H, FU Y. Establishment of a murine model of cerebral malaria in KunMing mice infected with *Plasmodium berghei* ANKA. *Parasitology.* 2016;143(12):1672-1680. doi:10.1017/S0031182016001475

Comment 6: *In adoptive transfer experiments, did the authors ensure that the neutrophils were not activated before delivery to recipient mice? How so?*

Answer to comment 6: We are sorry for the unclearness. For adoptive transfer, the method was in line with a previously published paper. In this publication¹, the mouse monocytes, macrophages, neutrophils, B cells and T cells from blood, spleen and liver homogenates were isolated by FACS with a FACS Aria II cell sorter. For adoptive transfer, splenocytes were incubated with 1 μ l anti-CD11b (clone M1/70, BD

Bioscience, San Jose, CA, USA) per 1×10^8 cells. Other publications also used same monoclonal antibodies to sort murine neutrophils for adoptive transfer experiment^{2,3}. We have revised method and added a new reference.

“Splenic neutrophils and macrophages were positively selected as previously described³². The purity of sorted cells, as verified by FACS analysis, was more than 95%, which separated intact live cells from dead cells and enucleated cellular debris. The proportion of dead cells was determined by using a hemocytometer to count the cells stained with trypan blue and was lower than 5%. Fifteen SOD3^{-/-} mice were randomly allocated to three groups with five animals in each group. In the SOD3^{-/-} + WT macrophage group, SOD3^{-/-} mice (recipients) received 1.5×10^6 macrophages from WT mice i.v. (SOD3⁺ macrophages→SOD3^{-/-} host). In the SOD3^{-/-} mice + WT neutrophils group, SOD3^{-/-} mice (recipients) received only 1.5×10^6 neutrophils from WT mice i.v. (SOD3⁺ neutrophils→SOD3^{-/-} host). SOD3^{-/-} mice and WT mice in the control group received the same volume of saline buffer. Mice were inoculated intraperitoneally with iRBCs.”

References:

1. Theurl, I., Hilgendorf, I., Nairz, M. et al. On-demand erythrocyte disposal and iron recycling requires transient macrophages in the liver. *Nat Med* 22, 945–951 (2016). <https://doi.org/10.1038/nm.414>
2. Jia, Yonghui et al. “Inositol 1,3,4,5-tetrakisphosphate negatively regulates phosphatidylinositol-3,4,5- trisphosphate signaling in neutrophils.” *Immunity* vol. 27,3 (2007): 453-67. doi:10.1016/j.immuni.2007.07.016
3. Prasad, Amit et al. “Inositol hexakisphosphate kinase 1 regulates neutrophil function in innate immunity by inhibiting phosphatidylinositol-(3,4,5)-trisphosphate signaling.” *Nature immunology* vol. 12,8 752-60. 19 Jun. 2011, doi:10.1038/ni.2052

Comment 7: *Correlation analyses and regression analysis are presented in the Results but are not described in Methods. This is important because the application of these tests is poorly described otherwise. In general, statistical analyses are very difficult to*

interpret in this manuscript and text descriptions of the data are difficult to follow. Parasitemia curves should not be compared day by day with t tests, as data are not independent and have a time component; the authors should consider area under the curve in these cases. A statistician should be consulted to review all of the data and analyses in this paper.

Answer to comment 7: We apologize for this unclearness. We have added Pearson correlation in method and reanalyzed the data and revised Figures.

Method: “Pearson correlation and regression analyses were used for the correlation studies. Correlation analysis was performed using the function “cor” of the base R package by applying Pearson’s method, using default parameters unless otherwise specified.”

Result: “Parasitemia was depicted with the area under curve (AUC). AUC = area under curve. k Infusion of recombinant IL-2 in WT mice inhibited *P. berghei* proliferation. Parasitemia was depicted with the AUC.”

Comment 8: *The survival curves shown in this work are the most exciting aspect of the study and are compelling.*

Answer to comment 8: We appreciate very much for the positive comment.

Comment 9: *In all cases where mice were immunologically manipulated (ablation experiments, for example), impacts on parasitemia should be shown and considered. This was done only in Fig 4 (IL-2 manipulation), albiet without showing a complete course of infection (graphs stop at days 9 and 11, yet some mice survived to day ~36).*

Answer to comment 9: We appreciate the suggestion very much. This manuscript included two immunologically manipulated experiments, the first was IFN- γ ablation and the second was IL-2 ablation. We only show the variation in parasitemia of the IL-2 experiments due to the fact that neutralizing IFN- γ cannot impact parasitemia in *Pb.ANKA* infected model. Other publications also showed the same results.

Additionally, we have had performed more experiments, only one mouse was survived to day 36, which was an outlier. Thus, we have removed this incorrect data in new

manuscript and the method section was revised.

Method: “The monitoring of parasitemia was quitted after the death of all animals in the control group.”

Reference

1. Villegas-Mendez, Ana et al. “IFN- γ -producing CD4⁺ T cells promote experimental cerebral malaria by modulating CD8⁺ T cell accumulation within the brain.” *Journal of immunology* vol. 189,2 (2012): 968-79. doi:10.4049/jimmunol.1200688

Comment 10: *None of the western blots shown are full blots, but are rather excessively trimmed. Full blots should be offered as supplemental data.*

Answer to comment 10: We agree with the reviewer. Full blots were added as Supplementary Figure 7.

Comment 11: *Throughout the Results section, immunological and other biochemical measurements are done very early in infection at time points that do not correspond to when mice succumb to infection. This makes it difficult to attribute the described changes to the observed outcomes. Examples include Fig 1i, Fig 2, Fig 3, Fig 4, and some of Fig 5.*

Answer to comment 11: We appreciate the comment. Studies have strongly suggested that an early burst of IFN- γ production is associated with protection to *P. falciparum* and *P. berghei* ANKA infection^{1,2}. Thus, this paper was focused early infection. the time points were in line with other publications. We think the change in late stage is out of scope of the current manuscript. We will address this in the future. Based on your comment, we have revised title and add discussion to highlight the scope of the current manuscript.

Title: SOD3 suppresses early cellular immune responses to parasite infection

Discussion: “Studies have strongly suggested that an early burst of IFN- γ production is associated with protection to *P. falciparum* and *P. berghei* ANKA infection^{23,24}.”

Discussion: “Moreover, neutralization of IL-2 at early time also led to decreased IFN- γ production in SOD3^{-/-} mice in response to parasite infection. Thus, the protective

effect of SOD3 deficiency is mediated by early production of IFN- γ .”

References

1. Mitchell, A.J., Hansen, A.M., Hee, L., Ball, H.J., Potter, S.M., Walker, J.C., and Hunt, N.H. Early cytokine production is associated with protection from murine cerebral malaria. *Infect Immun* 73, 5645-5653 (2005).
2. Vigário, A.M., Belnoue, E., Grüner, A.C., Mauduit, M., Kayibanda, M.I., Deschemin, J.-C., Marussig, M., Snounou, G., Mazier, D., Gresser, I., and Rénia, L. Recombinant Human IFN- α Inhibits Cerebral Malaria and Reduces Parasite Burden in Mice. *J Immunol* 178, 6416-6425 2007.

Comment 12: *Fig 1 does not make a compelling case that SOD3 is a strong determinant in outcomes in toxoplasmosis.*

Answer to comment 12: We apologize for this ambiguity. In this study, we used highly virulent *T. gondii* RH strain to infected SOD3^{-/-} mice and wild type mice. The result show that over 70% SOD3^{-/-} mice infected with *T. gondii* were still alive when the controls were died. Some paper reported a novel vaccine just delay survival time only one or two days compared to control in *T. gondii* RH infected model. We have removed unclear data to make sure our description is consistent with the **Figure 1g**.

Reference

1. Wu, Y., Ying, Z., Liu, J. et al. Depletion of Toxoplasma adenine nucleotide translocator leads to defects in mitochondrial morphology. *Parasites Vectors* 15, 185 (2022). <https://doi.org/10.1186/s13071-022-05295-7>
2. Tian X, Sun H, Wang M, Wan G, Xie T, Mei X, Zhang Z, Li X and Wang S (2022) A Novel Vaccine Candidate: Recombinant Toxoplasma gondii Perforin-Like Protein 2 Stimulates Partial Protective Immunity Against Toxoplasmosis. *Front. Vet. Sci.* 8:802250. doi: 10.3389/fvets.2021.802250

Comment 13: *Specific criticisms of data analysis: Fig 1b, t test is not appropriate for multiple comparisons against day 0; Fig 2g also needs correction for multiple hypothesis testing within the data; Fig 2a, Fig 3a and Fig 4l results are impossible to*

interpret as presented and described; Fig 1e should be presented as a full course of infection with comparison of area under the curve; sample sizes are not consistently provided in figure legends and time points at which data were collected are not provided

Answer to comment 13: We appreciate the suggestion very much. we Figures and legends have been revised accordingly.

In the previous manuscript we only showed parasitemia at one time point during *P. chaharani* infection (Fig 1e). This experimental result is only complementary to our results. Therefore, we have deleted this part of the data in the revised manuscript.

The statistics of Figure 1b have been changed to one-way variance test.

Figure legend: “Five biological replicates were included in the analysis; Uninfected controls were three biological replicates. Statistical significance tested by one-way ANOVA, and multiple comparisons to each day were corrected using Dunnet’s method. WT mice of average survival time was 9 days from the time of challenge.”

WT mice of average survival time was 9 days from the time of challenge.”

The statistics of Fig 2g corresponding to new Fig 2 b have been corrected by using multiple hypothesis testing within the data.

Comment 14: Fig2b and c don’t seem to be representing the same data (ie changes in expression level of SOD3 are unconvincing in the blot); densitometry for Fig 3h should be provided, and the interpretation of such data reviewed as the text does not match what is shown in the graph, especially for TLR7 and TLR8; Fig 4c-f need to be labeled to identify source of cells; Fig 5a-d need a legend to identify which groups are being

shown (and it is difficult to match what is described in the text to what is shown in these panels)

Answer to comment 14: We agree the comment very much. Fig2b and c have been deleted in the revised manuscript. The Fig 3h was moved to Supplementary Information. Fig 4c-f has been labeled. The Fig 5a-d have been revised.

Comment 15: *Throughout this manuscript, various timelines are chosen for representation in the figures without justification. For the most part, the immunologic changes shown are from early timepoints in the infections, yet survival curves show very extended times to death, in some cases weeks later. Thus, it becomes difficult to associate the immune changes with disease outcomes. While it is appreciated that WT mice succumb quickly to ECM and the authors attempted to compare events in those mice with those in null mice, the latter DO eventually succumb to infection (in all models shown except adoptive transfer in Supp Fig 3), and there is no examination whatsoever in the manuscript to explore what ultimately does induce death for these mice. Another important issue is that the null mice, by most criteria measured, are at baseline immunologically quite different than their WT counterparts. The authors are far as this reviewer can tell, never discuss this point clearly, nor consider the implications. The authors make some claims about causation and temporal relationships that are either not born out by the data or are not evident in the data as shown, for example lines 253-5, 288-9, 361-3.*

Answer to comment 15: We appreciate and agree the comment very much. In this study, we observed that the level of SOD3 was significantly elevated in both *Plasmodium falciparum* malaria patients and mice infected with five parasite species. SOD3-deficient mice had a substantially longer survival time and lower parasitemia than control mice after infection, whereas SOD3-overexpressing mice were much more vulnerable to parasite infection. We revealed that SOD3, secreted from activated neutrophils, preferentially bound to T cells, suppressed the interleukin-2 (IL-2) expression and concomitant interferon-gamma (IFN- γ) responses crucial for parasite clearance.

Experimentally, we have addressed the issue as follows:

- 1) To investigate whether SOD3 can directly bind and regulate immune cells, we conducted cell binding experiments using the recombinant SOD3 protein (rSOD3). Results showed that the rSOD3 directly bound to T cells, while its interaction with other cell types was much weaker (Fig. 3a).
- 2) We have isolated T cells and determined that SOD3 can directly inhibit the expansion of IFN- γ -producing T cell subpopulations. rSOD3 could directly suppress T cell activation (Fig. 3g) and inhibited naive T cell differentiation into effector T cells (Fig. 3h and i). Importantly, rSOD3 significantly reduced the proportion of IFN- γ -producing T cells in vitro, suggesting that SOD3 directly inhibited the expansion of IFN- γ -producing T cells (Fig. 3j).
- 3) Western blot shown that an increased expression of JNK was observed in purified T cells in SOD3^{-/-} mice after infection (Fig. 3k and l)
- 4) Inhibition JNK activity with a specific inhibitor SP600125 impaired proliferation of IFN- γ -producing T cells (Fig. 3m).
- 5) An increased proportion of IL-2-producing CD4⁺ T cells and CD8⁺ T cells was also observed in SOD3^{-/-} mice after *P. y. yoelii* infection (Fig. 4f-h).
- 6) Addition of SOD3 into the splenic lymphocytes significantly reduced the proportion of IL-2-producing CD4⁺ T cells and CD8⁺ T cells, suggesting that SOD3 can directly regulate the expansion of IL-2-expressing T cells (Fig. 5a and b). To investigate whether the inhibition of IL-2-expressing T cells by SOD3 is causally related to JNK activation, we assessed the proportion of IL-2-producing T cells under JNK-deficient conditions. Inhibition of JNK activity led to the reduction of IL-2-producing CD4⁺ and CD8⁺ T cells (Fig. 5c and d).
- 7) SOD3 deficiency significantly increased the proportion of IFN- γ expressing CD122⁺ (IL-2 receptor) T cells without altering the proportion of CD122⁺ T cells (Fig. 5e and f).

We hope that with this analysis we could adequately address the reviewer's concern, and the text of the manuscript has been expensively revised.

Reviewer #2

Comment 1: *Authors examined the role of SOD3, extracellular ROS scavenger, in Plasmodium infection and other protozoan infections. Authors showed that SOD3-deficient mice are resistant to Plasmodium infection, while SOD3-overexpressing mice are more vulnerable. Authors suggest that SOD3 regulate cytokine production by host immune cells and regulate host immunity during protozoan infection. Authors described experiments using multiple species of protozoan parasites in mice and in human malaria infection. However, each experiment is poorly described and lacks the critical information that justify their conclusion. I advise authors to focus this study on malaria alone and describe details of the methods.*

Answer to comment 1: We appreciate very much and agree with the comment. In the revised manuscript, we presented data mainly from Plasmodium infected animals, and more details of the methods were added in Supplementary Information.

Comment 2: *In addition, authors did not address how SOD3 activity is linked to the inhibition of cytokine production by T cells and NK cells, while they showed that cytokine production was improved in SOD3 KO mice in malaria infection model. Authors only mentioned that SOD3 could directly repress the activation of T cells (line 355).*

Answer to comment 2: Thank you very much for the comment, we have added new result to address how SOD3 activity is linked to the inhibition of cytokine production by T cells.

Result: “We have isolated T cells and determined that SOD3 can directly inhibit the expansion of IFN- γ -producing T cell subpopulations. rSOD3 could directly suppress T cell activation (Fig. 3g) and inhibited naive T cell differentiation into effector T cells (Fig. 3h and i). Importantly, rSOD3 significantly reduced the proportion of IFN- γ -producing T cells in vitro, suggesting that SOD3 directly inhibited the expansion of IFN- γ -producing T cells (Fig. 3j).”

Result: “Western blot shown that an increased expression of JNK was observed in purified T cells in SOD3^{-/-} mice after

infection (Fig. 3k and l). Inhibition JNK activity with a specific inhibitor SP600125 impaired proliferation of IFN- γ -producing T cells (Fig. 3m).”

Result: Addition of SOD3 into the splenic lymphocytes significantly reduced the proportion of IL-2-producing CD4⁺ T cells and CD8⁺ T cells, suggesting that SOD3 can directly regulate the expansion of IL-2-expressing T cells (Fig. 5a and b). To investigate whether the inhibition of IL-2-expressing T cells by SOD3 is causally related to JNK activation, we assessed the proportion of IL-2-producing T cells under JNK-deficient conditions. Inhibition of JNK activity led to the reduction of IL-2-producing CD4⁺ and

CD8⁺ T cells (Fig. 5c and d).

We hope that with this analysis we could adequately address the reviewer's concern.

Comment 3: *Introduction lacks background information of the study. What is known about SOC3 and what has been investigated? Why understanding of redox signaling that facilitate pathogen evasion of host immune response is challenging? What are known regarding SOD1 and SOD2?*

Answer to comment 3: We appreciate very much for the comment, we have extensively revised the **Introduction**.

“Host immune responses are tightly controlled by various immune factors during infection^{1,2}. Pathogens are known to hijack host factors³ or express an array of virulence factors⁴⁻⁷ that aim to overcome host immune defenses to achieve successful proliferation and dissemination. Host genetic and immune factors (HIFs) are the two major types of factors that are exploited by pathogens during infection^{8,9}. The reactive oxygen species (ROS) are one types of the innate responses upon pathogen invasion¹⁰. However, ROSs are not only detrimental to the invaded pathogens, but also harmful to the host cells. The host cells rely on the superoxide dismutases (SODs, including SOD1, SOD2 and SOD3) to scavenge the extra amount of ROS and the maintenance of the homeostasis. It is known that *Plasmodia* increased a substantial amount of SOD activity from host in the infection¹¹. However, a fundamental challenge in studying the HIF-pathogen interactions is to understand the host derive-redox signaling that facilitates pathogen evasion of host immune responses. Superoxide dismutase 3 (SOD3) is the only SOD-like enzyme secreted from the host cells to the extracellular space that scavenges substantial amounts of ROS to protect the host from oxidative stress during infection^{12,13}. Additionally, SOD1, primarily functions in the cytoplasm, has been reported to serve as an indicator of disease severity in individuals with various clinical manifestations of *vivax* malaria¹⁴. SOD2, mainly localized in the mitochondrial lumen, has been found to be notably upregulated in pre-asymptomatic malaria patients from Cameroon¹⁵. Although the roles of SOD1 and SOD2 in malaria are well characterized^{14,15}, almost nothing is known about the contribution of SOD3 to host immune responses

beyond ROS scavenging.

Here, we demonstrated that SOD3 mainly secreted by neutrophils could directly bind T cells and suppress its IL-2 production, and consequently reduced the recruitment of IFN- γ - producing T cells in the responses to invading parasites. Our results revealed a novel, but essential, role of SOD3 in responses to parasite invasion and targeting SOD3 may be an alternative way to enhance host protective responses and reduce disease severity in acute infection.”

Comment 4: *References of the experimental animals used in this study should be indicated.*

Answer to comment 4: We thank the reviewer for this great comment and have included all the information in the **Method section**. In the previous submission, the reason for not including the detail references of the experimental animals was the requirement of “Double-blind peer review”. This has been revised in the resubmission. **Method:** “Female and male C57BL/6 mice were obtained from Liaoning Changsheng Biological Technology Company (Liaoning, China).”

Comment 5: *Information of human studies should be described in much more in detail. How were malaria patients recruited? What was the criteria of selecting 20 patients? What are the names of the hospitals? What are age and sex of these patients and healthy volunteers?*

Answer to comment 5: We thank the reviewer for this great comment and have included all the data in the **Method** section. In the previous submission, the reason for not including the detail information of the malaria patients was the requirement of “Double-blind peer review”. This has been revised in the resubmission.

“All procedures performed on human samples were carried out in accordance with the tenets of the World Medical Association's Declaration of Helsinki. A total of 20 patients (ten male and ten female) suffering from *falciparum* malaria were recruited from a malaria-endemic area, all patients provided with written informed consent as previously described. Another batch of serum samples were collected from *P. falciparum*-infected

patients (24 male) with a recent history of travel to malaria-endemic African countries, and *P. falciparum* infection was clinically confirmed by a rapid diagnostic test. The inclusion criteria were no history of *P. falciparum* infection. Healthy control sera were collected from healthy volunteers with informed consent. The information of all individuals involved was anonymized. Written consent for the publication of this study was obtained from all individuals. Studies in humans were reviewed and approved by the Ethical Committee of the Chinese Academy of Medical Sciences (approval no. IPB-2016-2) and Institutional Review Board (no. PRAMS0034319) of the Pennsylvania State University.”

Comment 6: *Information of parasites used in the study should be described in much more detail and references should be indicated. Write full name of each parasite and how authors obtained these materials.*

Answer to comment 6: We are sorry for the confusion, as requested for the reason of “double-blind” submission, all information may indicate the authorship was removed in the previous manuscript. Now, all detailed information of the parasite lines have been added in the **Method** section:

“*Plasmodium* species lethal to rodents, including *P. y. yoelii* YM, *P. y. yoelii* 17XL and *P. berghei* ANKA, were donated by Dr. Yaming Cao and propagated by passage in C57BL/6 male mice of 7–8 weeks old; a transgenic *P. berghei* ANKA parasite strain expressing luciferase under the control of the *efl1a* promoter (PbA luc) was propagated by passage in mice. ”

Comment 7: *The amount of monoclonal antibody used for in vivo is too low for neutralization.*

Answer to comment 7: We apologize for this error; We used 10 µg/g to neutralize IL-2 for *in vivo*. The method section has been revised.

Comment 8: *Methods used for SOD3 detection should be explained much more in detail.*

Answer to comment 8: We thank the reviewer for this the comment and have made revision in the **Method** section:

“The concentrations of SOD3 in the sera of WT mice at different time points post-infection were detected using a mouse SOD3 ELISA kit (CUSABIO, Wuhan, China, Cat# CSB-EL022399MO) according to the manufacturer’s instructions. Briefly, 100 μ L of different concentrations of standards and samples were added to each well. The wells were then covered with an adhesive strip and incubated for 2 hours at 37°C. After removing the liquid from each well, 100 μ L of biotin antibody (1x) was added to each well after three washes with PBS. The wells were covered with a new adhesive strip and incubated for 1 hour at 37°C. Next, each well was aspirated and washed three times. After washing, 100 μ L of HRP-avidin (1x) was added to each well and incubated for 1 hour at 37°C, followed by five washes. Finally, after the addition of TMB substrate, the optical density (OD) at 450 nm was measured. The detection range of is 7.8 pg/mL-500 pg/mL.”

Comment 9: *Parasitemia should be determined throughout the course of the infection with parasites, not just one time-point.*

Answer to comment 9: We thank the reviewer for this great comment. In the previous manuscript we only showed parasitemia at one time point during *P. chaharani* infection. This experimental result is only complementary to our results. Therefore, we have deleted this part of the data in the revised manuscript.

Comment 10: *Approximately 50% of SOD3 KO mice died in 14 days after P. berghei infection. What was the parasitemia levels? Why did these mice died without parasite accumulation in the brain (Fig. 2d)?*

Answer to comment 10: We appreciate the question. SOD3-deficient mice had a substantially longer survival time and lower parasitemia than control mice after infection at the same time after infection. The mice in control group were died in day 9.

The detection time in Figure 2d is not the time of death. During the same infection time, SOD3-deficient mice shown lower parasite accumulation in the brain than WT and overexpressed-mice.

Comment 11: For all flow cytometry analysis, show gating strategy and representative flow cytometry profiles.

Answer to comment 11: We thank the reviewer for this great comment. The gating strategy and representative flow cytometry profiles have been shown.

Comment 12: Authors should also examine *Plasmodium* antigen-specific cytokine production by T cells.

Answer to comment 12: We thank the reviewer for this great comment. But we think this is out of scope of the current manuscript. We have added Figure 3, 4, and 5 to prove the relation between SOD3 and cytokine production. We will address *Plasmodium* antigen-specific cytokine production by T cells in the future.

Comment 13: Show reference of single-cell transcriptome data of SOD3 expression (line 290-292).

Answer to comment 13: We appreciate comment and have made revision in **Result** section.

“To identify the source of splenic SOD3, we analyzed available single-cell transcriptome data (Mouse Cell Atlas, <https://bis.zju.edu.cn/MCA/>) from healthy mice.”

Comment 14: *Explain the experiment in Fig. 5i in more detail. Did authors prepare macrophages and neutrophils from uninfected mice? Are they from spleen? Did they transfer to infected mice? What day after infection? What was the purity?*

Answer to comment 14: We are sorry for the unclearness. The splenic macrophages and neutrophils was sorted from uninfected WT mice and then transfer to uninfected SOD3^{-/-} mice. Meanwhile, these recipients and negative control mice were infected with iRBCs.

“Splenic neutrophils and macrophages were positively selected as previously described³². The purity of sorted cells, as verified by FACS analysis, was more than 95%, which separated intact live cells from dead cells and enucleated cellular debris. The proportion of dead cells was determined by using a hemocytometer to count the cells stained with trypan blue and was lower than 5%. Fifteen SOD3^{-/-} mice were randomly allocated to three groups with five animals in each group. In the SOD3^{-/-} + WT macrophage group, SOD3^{-/-} mice (recipients) received 1.5×10^6 macrophages from WT mice i.v. (SOD3⁺ macrophages→SOD3^{-/-} host). In the SOD3^{-/-} mice + WT neutrophils group, SOD3^{-/-} mice (recipients) received only 1.5×10^6 neutrophils from WT mice i.v. (SOD3⁺ neutrophils→SOD3^{-/-} host). SOD3^{-/-} mice and WT mice in the control group received the same volume of saline buffer. Meanwhile, mice in all groups were inoculated intraperitoneally with iRBCs.”

REVIEWER COMMENTS

Reviewer #1 (Remarks to the Author):

The authors have attempted to be responsive to all three peer reviews. Extensive changes have been made in acknowledgement of the concerns.

Reviewer #2 (Remarks to the Author):

The manuscript was improved from the original version but is still descriptive and lacks mechanistical insights. More importantly, authors are not successful in explaining mechanisms underlying the upregulation of the immune responses in the absence of SOD3 and misinterpret some of their data as I explain in more detail. In addition, some of the methods are still not clear and references are missing.

Major points,

1. SOD3 has enzymatic activity of extracellular superoxide dismutase and has been described as an antioxidant immobilized in the extracellular space. It is crucial to determine whether immunoregulatory property of SOD3 in mice infected with malaria parasites depends on its enzymatic activity or is mediated by other function of SOD3 independent on its enzymatic activity.
2. Authors showed that SOD3 bind CD4+ T cells but poorly to CD8+ T cells and NK cells (Fig. 3a). However, their immunological analysis of CD4, CD8 and NK cells show that all of them exhibit enhanced response in SOD3 KO mice when compared with WT mice. These data indicate that there is no correlation between binding of SOD3 and T cell function. Authors misinterpreted these data in several portions in the manuscript.

Other points,

3. Reference of SOD3 KO mice is missing (line 94, Method). Did authors generate these mice for this study?
4. Reference of luciferase expressing P. berghei ANKA and SOD3 transgenic mice are missing (line 141, 142, Method).
5. In Fig. 2e, SOD3 KO mouse control group is missing. It is critical to have control group in the same experiment.
6. Production of IFN-g by CD4, CD8 and NK cells, should be examined separately to address correlation between SOD3-binding and cellular function. Only CD4 T cells bind strongly to SOD3 (Fig. 3A).
7. How authors interpret the increase in IFN-g production by SOD3 KO T cells prior to the infection (time 0, Fig. 3b).
8. JNK expression in supplementary Fig. 4 and Fig. 3k is contradictory. It seems no difference in the former and upregulated in the latter, SOD3 KO in Fig.3k.
9. Method of detecting bound SOD3 should be described in more detail (line 218, Method). What secondary antibodies did authors used?
10. Authors mentioned that SOD3 preferentially binds CD4 T cells and CD8 T cells (Fig. 3a)(line 264). This is clearly wrong statement as I mentioned previously.
11. Authors have not shown sufficient evidence that inhibitory effect of SOD3 on T cells is mediated by JNK (line 273, 341). How do authors explain mechanisms underlying the inhibition of JNK?

Response to the reviewer's comments:

We thank the reviewers for finding the revised manuscript suitable for publication at *Nature Communications*. Below is a point-by-point response to the remaining concerns of reviewer #2 with our responses shown in blue.

Reviewer #2 (Remarks to the Author):

Comment 1: The manuscript was improved from the original version but is still descriptive and lacks mechanistical insights. More importantly, authors are not successful in explaining mechanisms underlying the upregulation of the immune responses in the absence of SOD3 and misinterpret some of their data as I explain in more detail. In addition, some of the methods are still not clear and references are missing.

Answer to comment 1: We thank the reviewer for the positive and constructive comment on our study. Regarding the mechanisms underlying the upregulation of the immune responses in the absence of SOD3, we propose that SOD3 negatively regulatory factor in the infected hosts, this was further supported by the data obtained from the *in vitro* experiments, where the function of T cells was inhibited after incubation with SOD3. The text in the **Methods**, **Results** and **Discussion** have been revised accordingly.

Comment 2: SOD3 has enzymatic activity of extracellular superoxide dismutase and has been described as an antioxidant immobilized in the extracellular space. It is crucial to determine whether immunoregulatory property of SOD3 in mice infected with malaria parasites depends on its enzymatic activity or is mediated by other function of SOD3 independent on its enzymatic activity.

Answer to comment 2:

We appreciate and agree with the comment. As the reviewer mentioned, SOD3 is an extracellular enzyme capable of modulating extracellular redox conditions by catalyzing the dismutation of superoxide into hydrogen peroxide. Consistent with prior research, we utilized active SOD3 (with a specific enzymatic activity of 273.8 U/mg) to explore its role in mice infected with *Plasmodium* parasites and *in vitro*. Moreover, the active SOD3 protein has been shown to exert a direct inhibitory effect on T cell function *in vitro*. Administering active SOD3 protein to SOD3 knockout mice infected with *Plasmodium* resulted in a mortality rate similar to that of wild-type (WT) mice post-infection. We acknowledge that the immunoregulatory property of SOD3 can also be independent of its enzymatic activity, as it indeed inhibits the production of IL-2 from the T cells *in vitro*. Further, we have now incorporated in the

Discussion a paragraph about the limitations of our study as followed:

While we believe that these results presented in this study is an advance in understanding the interaction between the invade protozoan parasites and host immune system, the exact mechanism underlies the possible regulation of SOD3 on the T cells remains further investigation. The reason is that, even though the active SOD3 protein showed direct inhibition on T cell function in both in vitro and in vivo conditions, other functions of SOD3 may mediate the effects independent of its enzymatic activity and affinity to the immune cells. Further, a significant increase in JNK expression was indeed observed in T cells sorted from SOD3^{-/-} mice after infection, there may be an unknown link between SOD3 and the kinase activity of the JNK protein.

Comment 3: Authors showed that SOD3 bind CD4⁺ T cells but poorly to CD8⁺ T cells and NK cells (Fig. 3a). However, their immunological analysis of CD4, CD8 and NK cells show that all of them exhibit enhanced response in SOD3 KO mice when compared with WT mice. These data indicate that there is no correlation between binding of SOD3 and T cell function. Authors misinterpreted these data in several portions in the manuscript.

Answer to comment 3:

We appreciate the reviewer's observation regarding the variable binding affinity of SOD3 among CD4⁺ T cells, CD8⁺ T cells, and NK cells, as illustrated in Figure 3a. The data are evident that SOD3 exhibits a preferential binding to CD4⁺ T cells, with comparatively weaker interactions observed with CD8⁺ T and NK cells. Despite these differences, a crucial point is that SOD3 is capable to bind both T and NK cells. This observation underpins our hypothesis that SOD3 may possess immunomodulatory roles. Nonetheless, it is important to acknowledge that our current data do not delve into how SOD3's binding affinity correlates with T cell functionality. As such, we propose that the potential of SOD3 to modulate T cell and NK cell responses might not be strictly dependent on its binding affinity on these cells. We have softened the link between the binding affinity and T cell function in this manuscript, as follows:

- (a) **Result:** “Results showed that the rSOD3 directly bound to T cells and NK cells (Fig. 3a).”
- (b) **Result:** “As SOD3 can bind CD4⁺ T cells and CD8⁺ T cells (Fig. 3a), we therefore tested whether SOD3 inhibits T cells to generate IL-2.”
- (c) **Discussion:** “While we believe that these results presented in this study is an advance in understanding the interaction between the invade protozoan parasites and host immune system, the exact mechanism underlies the possible regulation of SOD3 on the T cells remains further investigation. The reason is that, even though the active SOD3 protein showed direct inhibition on T cell function in both in

vitro and in vivo conditions, other functions of SOD3 may mediate the effects independent of its enzymatic activity and affinity to the immune cells. Further, a significant increase in JNK expression was indeed observed in T cells sorted from SOD3^{-/-} mice after infection, there may be an unknown link between SOD3 and the kinase activity of the JNK protein.”

Comment 4: Reference of SOD3 KO mice is missing (line 94, Method). Did authors generate these mice for this study?

Answer to comment 4: We are sorry for the confusion. SOD3 KO mice were obtained from Cyagen Biosciences. We have revised the text in the Result and Method sections.

Result (line 94): To test this hypothesis, we used SOD3 knockout (SOD3^{-/-}) mice.

Method: C57BL/6-Sod3tm1cyagen (SOD3^{-/-}) mouse strains (Serial Number: KOCMP-22050-Sod3) were purchased from Cyagen Biosciences (Suzhou, China).

Comment 5: Reference of luciferase expressing *P. berghei* ANKA and SOD3 transgenic mice are missing (line 141, 142, Method).

Answer to comment 5: The reference of luciferase expressing *P. berghei* ANKA and SOD3 transgenic mice has been added it.

Method: The reference (ref 28) regarding the transgenic *P. berghei* ANKA parasite strain expressing luciferase under the control of the efla promoter (*PbA luc*)¹ was added.

Method: SOD3 transgenic mice (Global SOD3 overexpression) were constructed with the help from GRENSTER Co., Ltd. (Liaoning, China).

Comment 6: In Fig. 2e, SOD3 KO mouse control group is missing. It is critical to have control group in the same experiment.

Answer to comment 6: We appreciate and agree with the comment. The data of a control group was now added in Fig. 2e.

Comment 7: Production of IFN-g by CD4, CD8 and NK cells, should be examined separately to address correlation between SOD3-binding and cellular function. Only CD4 T cells bind strongly to SOD3 (Fig. 3A).

Answer to comment 7: We are grateful for the instructive comment, which has prompted us to further explore the functional implications of SOD3 binding to T cells and NK cells. As suggested by the reviewer, we have conducted additional experiments, both *in vitro* and *in vivo*, to investigate the effect of SOD3 on IFN- γ production in CD4⁺ T cells. Our findings indicate that the absence of SOD3 in the knockout mice leads to a significant increase in IFN- γ -producing CD4⁺ T cells following infection, compared to wild-type (WT) mice (as shown in Figure a). This observation is complemented by *in vitro* experiments showing that SOD3 treatment can inhibit IFN- γ production in CD4⁺ T cells (as shown in Figure b). The data further

support our proposal that SOD3 acted as a negative regulator to the T cells. The results gained from these additional experiments not only address the initial limitation identified by the reviewer but also open new avenues for future research to unravel the mechanistic

underpinnings of SOD3's immunomodulatory functions across different lymphocyte populations, with a particular focus on its binding affinities. We have now incorporated in the **Discussion** a paragraph about the limitations of our study.

Discussion: “While we believe that these results presented in this study is an advance in understanding the interaction between the invade protozoan parasites and host immune system, the exact mechanism underlies the possible regulation of SOD3 on the T cells remains further investigation. The reason is that, even though the active SOD3 protein showed direct inhibition on T cell function in both *in vitro* and *in vivo* conditions, other functions of SOD3 may mediate the effects independent of its enzymatic activity and affinity to the immune cells. Further, a significant increase in JNK expression was indeed observed in T cells sorted from SOD3^{-/-} mice after infection, there may be an unknown link between SOD3 and the kinase activity of the JNK protein.”

Comment 8: How authors interpret the increase in IFN-g production by SOD3 KO T cells prior to the infection (time 0, Fig. 3b).

Answer to comment 8: We are sorry for the confusion. The elevated levels of IFN- γ were detected in serum samples, not directly from SOD3 KO T cells (as indicated in Figure 3b). Importantly, the observation of increased serum IFN- γ in SOD3 KO mice even prior to infection not only corroborates our findings but also emphasizes the role of SOD3 in inhibiting IFN- γ production in healthy conditions. This suggested that

SOD3 might exert its immunomodulatory effects through a common mechanism that negatively influences IFN- γ production across different states of mice. We have made the necessary revisions to the **Figure legend** of Figure 3b to more accurately reflect these findings. This comment helps to clarify the impact of SOD3 on IFN- γ levels in both conditions and further supports our proposal of SOD3's broad regulatory role in immune modulation.

Figure legend: Depletion of SOD3 in mice rescued the IFN- γ production in the sera. Upregulated serum IFN- γ levels observed in SOD3 knockout mice in infection and healthy conditions.

Comment 9: JNK expression in supplementary Fig. 4 and Fig. 3k is contradictory. It seems no difference in the former and upregulated in the latter, SOD3 KO in Fig.3k.

Answer to comment 9: We apologize for the confusion. Supplementary Fig. 4 and Fig. 3k depict quantifications of JNK expression in different cell types. Specifically, in Supplementary Fig. 4, JNK expression was quantified by Western blotting in splenic cells, showing slightly increased expression in spleens of mice after 8 days. In contrast, in Fig. 3k, JNK expression was quantified in purified T cells, which aligns with the previous result.

Comment 10: Method of detecting bound SOD3 should be described in more detail (line 218, Method). What secondary antibodies did authors used?

Answer to comment 10: We appreciate the comment. Detailed description on detection of SOD3 on the cells was added in the method section, and the secondary antibody is The Alexa Fluor™ 488-conjugated goat anti-Rabbit IgG (H+L). The text of **Supplementary Method** and **Figure legend** have been revised.

Supplementary Method: His-SOD3 fusion protein was incubated with spleen immune cells as previously described. The SOD3 bound to the cell surface was detected by flow cytometry using anti-SOD3- Rabbit antibody (Affinity, DF7753) and Alexa Fluor™ 488-conjugated goat anti-Rabbit IgG (H+L) Highly Cross-Adsorbed Secondary Antibody (Invitrogen, A-11034). Flow cytometric analyses were performed using EV450 -conjugated anti-mouse CD45 (Elabscience, E-AB-F1136Q), Percp cy5.5-conjugated anti-mouse CD4 (Biolegend, 100433), APC-conjugated anti-mouse CD8a (Biolegend, 100711), PE-conjugated anti-mouse NK1.1 (Elabscience, E-AB-F0987D).

Figure legend: The Alexa Fluor™ 488-conjugated goat anti-Rabbit IgG (H+L) secondary antibody was used to recognize the primary antibody.

Comment 11: Authors mentioned that SOD3 preferentially binds CD4 T cells and CD8 T cells (Fig. 3a) (line 264). This is clearly wrong statement as I mentioned

previously.

Answer to comment 11: We appreciate and agree the comment. We have deleted the word “preferentially” and softened the link between SOD3 affinity and T cell function.

Comment 12: Authors have not shown sufficient evidence that inhibitory effect of SOD3 on T cells is mediated by JNK (line 273, 341). How do authors explain mechanisms underlying the inhibition of JNK?

Answer to comment 12: We are sorry for the confusion. We have softened the statement regarding the inhibitory effect of SOD3 on T cells is mediated by JNK throughout the manuscript. We have now incorporated in the **Discussion** a paragraph about the limitations of our study: While we believe that these results presented in this study is an advance in understanding the interaction between the invade protozoan parasites and host immune system, the exact mechanism underlies the possible regulation of SOD3 on the T cells remains further investigation. The reason is that, even though the active SOD3 protein showed direct inhibition on T cell function in both in vitro and in vivo conditions, other functions of SOD3 may mediate the effects independent of its enzymatic activity and affinity to the immune cells. Further, a significant increase in JNK expression was indeed observed in T cells sorted from SOD3^{-/-} mice after infection, there may be an unknown link between SOD3 and the kinase activity of the JNK protein.

REVIEWER COMMENTS

Reviewer #2 (Remarks to the Author):

I am satisfied with most of the author's response. However, I still have problem in the staining of SOS3 on T and NK cells (Fig. 3a). Authors showed control staining with AF488-Goat anti-Rabbit IgG secondary antibody alone. This is not right control. Control staining should be isotype control of the primary antibody and AF488-goat anti-Rabbit IgG secondary antibody. It is important, in particular, to justify the positive staining of CD8 T and NK cells.

Response to the reviewer's comments:

We are very grateful for your affirmation of our revised manuscript. Below is a point-by-point response to the last remaining issue of reviewer #2 with our responses shown in blue.

Reviewer #2 (Remarks to the Author):

Comment 1: Reviewer #2 (Remarks to the Author):

I am satisfied with most of the author's response. However, I still have problem in the staining of SOD3 on T and NK cells (Fig. 3a). Authors showed control staining with AF488-Goat anti-Rabbit IgG secondary antibody alone. This is not right control. Control staining should be isotype control of the primary antibody and AF488-goat anti-Rabbit IgG secondary antibody. It is important, in particular, to justify the positive staining of CD8 T and NK cells.

Answer to comment 1: We thank the reviewer for the positive and constructive comment on our study. We have added a control group as per your comment. The primary antibody used in this study is a rabbit-derived polyclonal anti-SOD3 antibody. Consequently, a non-specific rabbit IgG was employed as the primary antibody in the control group. The control antibody did not show any reaction with the cells, while the SOD3-specific antibodies did recognize the binding between SOD3 and CD8 T cells and NK cells. The **Figure 3a** and the **Figure legend**, as well as the text in the **Supplementary Method** have been revised accordingly.

Figure legend: “Control sample was stained with a non-specific rabbit IgG as the primary antibody and AF488-goat anti-rabbit IgG secondary antibody.”

Supplementary Method: “For the control staining, cells were treated with a non-specific rabbit IgG as the primary antibody, using the same AF488-conjugated goat anti-rabbit IgG as the secondary antibody.”

REVIEWERS' COMMENTS

Reviewer #2 (Remarks to the Author):

I am satisfied with the changes made by authors.